# Understanding Psychosocial Barriers to Healthcare Technology Adoption: A Review of TAM Technology Acceptance Model and Unified Theory of Acceptance and Use of Technology and UTAUT Frameworks

**DOI:** 10.3390/healthcare13030250

**Published:** 2025-01-27

**Authors:** Ann Thong Lee, R Kanesaraj Ramasamy, Anusuyah Subbarao

**Affiliations:** 1Faculty of Management, Multimedia University, Cyberjaya 63100, Malaysia; lee.ann.thong@student.mmu.edu.my (A.T.L.); anusuyah.subbarao@mmu.edu.my (A.S.); 2Faculty Computing Informatics, Multimedia University, Cyberjaya 63100, Malaysia

**Keywords:** Technology Acceptance Model (TAM), Unified Theory of Acceptance and Use of Technology (UTAUT), healthcare

## Abstract

**Background:** Leveraging healthcare technology improves human development and well-being. However, adoption is frequently delayed by behavioural and psychological barriers, such as perceived usefulness, trust, and organisational readiness. This review examines the suitability of the Unified Theory of Acceptance and Use of Technology (UTAUT) and the Technology Acceptance Model (TAM) frameworks in healthcare settings, focusing on behavioural, educational, and psychological factors that influence technology adoption. **Methods:** A total of 20 peer-reviewed articles from 2019 to 2024 were examined. **Results:** The review identified significant organisational and psychological obstacles, including a lack of trust, inadequate training, and organisational support. While the UTAUT provided a more comprehensive viewpoint, it needed to be modified to include context-specific factors, including trust, facilitating circumstances, and educational interventions. Meanwhile, the TAM’s emphasis on perceived usefulness and ease of use was shown to be insufficient for dealing with complex healthcare situations. **Conclusions:** Interventions targeting stakeholders’ organisational and psychological preparation and educational strategies are essential to overcoming resistance and enhancing trust. Future research should look into integrative frameworks incorporating behavioural, psychological, and instructional tactics to improve the use of technology in healthcare.

## 1. Introduction

Technologies were have been employed in the current century to improve the quality of life for of humans. Thus, technological devices and artificial intelligence (AI) were have been developed to assist humans in daily repetitive daily activities to ease the burden and save time. Artificial intelligence (AI) and electronic devices rely primarily on AI algorithms as their foundation to carry out all their jobs independently. As a result, these devices have been allowed to replace low-efficiency human workers in repetitive jobs such as labourers on assembly lines and manufacturing lines, among others.

With the advancement of technology, electronic devices and artificial intelligence (AI) could replace human labour in some positions that require highly accurate output and good time management abilities. For example, public transportation, such as trains, was is mainly used by people as their preferred method of transportation since it was is convenient and practical. Thus, the trains needed to arrive on schedule. By replacing the human workers with an electrical device and using artificial intelligence (AI), it was is possible to reduce the number of drivers and thus lower the cost of hiring them while also increasing the system’s effectiveness, for example, by ensuring that passengers arrived at their destinations on time or within as scheduled.

Technology has made life easier and decreased the likelihood of human error, which is one of the most well-known advantages it has brought to society. As is widely known, the healthcare industry has long struggled with a labour shortage [1,2]. When human beings have to, dealing with multiple issues at once, it could result in mistakes since it is difficult for people to do carry out numerous tasks at once. Human mistakes could lead to serious problems, especially in the healthcare industry. Even a tiny mistake in the healthcare field could have serious consequences, even death. Errors might have included missing something important, misdiagnosing someone, using the wrong therapy because of insufficient knowledge, and more others [2]. Thus, technology could have been employed to help reduce these issues [2].

Although technology has become more and more commonplace in this age of technological innovation, practitioners in the healthcare sector, in particular, continue to lack faith in technological innovation and artificial intelligence (AI) [3]. This has resulted in a lack of trust in the healthcare industry to embrace innovation, even if it has the potential to revolutionise services and the results of patient treatment. This suspicion stems from concerns about the potential for new technology to make significant mistakes, perhaps resulting in fatalities. As a result, it is still difficult for practitioners in the healthcare industry to trust technology. They tended to believe more in their skills and considered medical technology unreliable and imperfect [2]. This implies that mistakes, or even patient deaths, could occur, and even patient deaths could result, which could put them in trouble because they recommended using the technology on the patient. For instance, a minor error during surgery, such as a chemical overdose, could result in quite serious consequences for them. As a real-world illustration, delays in the implementation of clinical decision support systems during the COVID-19 pandemic resulted efficiencies and missed opportunities to enhance patient care. This situation underscores the importance of removing barriers that hinder the adoption of healthcare technologies.

Numerous studies have been conducted to help better understand better the variables influencing technology adoption in the healthcare sector. Different theoretical frameworks have been employed to identify the factors in this type of study. The two most popular theories are the Technology Acceptance Model (TAM) and the Unified Theory of Acceptance and Use of Technology (UTAUT), which are the essential theoretical research models for assessing the level of technology adoption [4,5]. Although the TAM and UTAUT are distinct theories, they aim to educate people about the deeper reasons behind their favourable or unfavourable opinions of the technology they are considering and how technology design can increase acceptance [6]. In order to gain a deeper comprehension of technological adoptions, numerous studies in the healthcare field have used the popular TAM and UTAUT frameworks [6]. However, several investigations have found that these models could not consistently predict medical technology acceptance and use [6]. This is because the TAM and UTAUT have significant limitations when employed in healthcare settings [6]. Specifically, these two models frequently overlook essential factors, including cultural diversity, trust, and domain-specific challenges, such as clinical decision-making procedures and regulatory restrictions. Therefore, bridging these gaps is imperative to gain a more comprehensive grasp of technology adoption in healthcare settings.

The TAM is a model that has been simplified and emphasises perceived usefulness and perceived ease of use, which has led to the model missing numerous complex variables that affect technology adoption in the healthcare industry [7]. On the other hand, it disregards several critical external factors that are required in the sector, like social effects, organisational support, and so on. Furthermore, the TAM believes that behavioural intention always leads straight to actual usage when, in actuality, there may be practical barriers, such as insufficient training for physicians and nurses in using new technology. Aside from that, the TAM is based on the Theory of Reasoned Action (TRA), which is a well-known model in psychology theory [8,9]. As a result, the TAM’s key components, its perceived ease of use and perceived usefulness, reflect a psychological perspective that directly influences an individual’s behavioural intentions.

Conversely, the UTAUT is more extensive, yet it might be challenging to apply in the healthcare setting due to its complexity. Due to its numerous components and moderating factors, the model may be challenging to implement and require large amounts of data collection, which may not be achievable in healthcare settings. Although the UTAUT aims for broad application, it can not take the particular complexities of healthcare settings into account, such as clinical decision-making processes and regulatory requirements [6]. Furthermore, the UTAUT may not sufficiently differentiate between the many roles and user types prevalent in the healthcare industry, such as doctors and nurses, each with a distinct interface for utilising technology. Apart from that, the UTAUT is also one of the most frequently used theories about technology or and user psychology that is most frequently used [10,11]. The model comprises four constructs representing an individual’s psychological perception of technology, among which the performance expectations and effort expectations are comparable to the model’s perceived usefulness and perceived ease of use, representing an individual’s psychological perception of technology. At the same time, the social influence and facilitating conditions reflect the role of social and organisational, as well as technological, factors.

Nevertheless, despite these limitations, these two models have strong theoretical foundations and demonstrated efficacy. Consequently, the majority of academics continue to concur that by including more variables, they might offer robust theoretical frameworks for a complex environment like a hospital setting. For example, several studies have extended the TAM and UTAUT to incorporate factors including user experience, organisational support, and trust. For instance, including privacy and trust considerations has improved the TAM’s ability to anticipate the adoption of teleconsultations [5,12,13]. Similarly, the UTAUT has been adjusted to meet the specialised requirements of healthcare professionals, incorporating components like performance standards and digital literacy, especially in environments with limited resources [14,15]. These modifications are designed to rectify model flaws and offer a more comprehensive picture of healthcare technology utilisation. By using these theories, researchers can pinpoint the factors that influence healthcare professionals’ adoption of the technology. This could improve the adoption of technological innovations in the healthcare industry and help healthcare professionals become more trusting of technology. However, even though both the TAM and UTAUT theories are appropriate for research about healthcare, researchers must take into account the context of the areas in which their study will take place as well as the potential scope of the investigation before deciding whether to use the TAM or UTAUT theory in their work, given the distinctions between the aspects and factors that these two theories emphasise. Comparing the TAM and UTAUT theories can help researchers better understand the advantages and limitations of each theory. They can then use the theory that best fits their research design to produce reliable study results, improve adoption strategies for new technologies, and increase the uptake and application of these technologies in the healthcare industry.

As a result, the objective of this study is to address the limits of current models by critically analysing the adaptability of TAM and UTAUT in the healthcare domain, identifying gaps in their application, and offering comprehensive strategies to solve domain-specific challenges, such as the psychosocial barriers within the healthcare settings. This study compiled and analysed the differences, advantages, and disadvantages of multiple research papers using different theoretical frameworks to investigate technology acceptance in the healthcare industry.

For this study, a total of two research questions are established:What are the key psychosocial and organisational factors influencing healthcare technology adoption as identified through studies using the TAM and UTAUT frameworks?How do psychological factors like trust and perceived usefulness impact healthcare professionals’ and patients’ adoption of technology?

## 2. Literature Review

As technology advances, work patterns have evolved to integrate these innovations, making life easier. Technological improvements have brought about changes in most fields. One excellent example of how technology has affected lives is the existence of automobiles [16]. Due to the novelty of technology, most people encounter difficulties adapting to the technological environment. As a result, numerous studies have been conducted to identify the factors that influence technology acceptability and assist in enhancing technological acceptance in society.

The healthcare industry faces one of the most serious problems in terms of technology adoption because people find it very difficult to adapt to new technologies despite the industry being the most important for people’s health [17]. This is because several factors influence the acceptance of such innovations. Numerous studies have been conducted to analyse the factors that cause barriers to technology acceptance in the healthcare industry. This study focuses on the two most commonly used theoretical frameworks in the healthcare industry: the Technology Acceptance Model (TAM) and the Unified Theory of Acceptance and Use of Technology (UTAUT).

The studies reviewed in this paper used either TAM theory or UTAUT theory as their theoretical frameworks, both of which are theories reflecting different perspectives, techniques, and conclusions. Nonetheless, each study aimed to encourage the acceptance of technological innovations in the healthcare industry for the benefit of society. Despite the widespread use of the TAM and UTAUT in healthcare-related studies, there is a lack of research scrutinising the benefits, constraints, and differences in these theories in healthcare-related contexts. Thus, this study will gather numerous healthcare-related studies, employing either the TAM or UTAUT theories, and use them as data in this study to analyse and determine the most suitable theoretical framework for technology adoption in the healthcare industry, as well as the advantages, limitations, and distinctions of the TAM and UTAUT theories.

### 2.1. Theoretical Framework

#### 2.1.1. The Technology Acceptance Model (TAM)

Numerous studies have indicated that the Technology Acceptance Model (TAM) is one of the most frequently used theories in the healthcare industry [5]. This theory aims to forecast how new information systems and technologies will be used and accepted by society while identifying the factors affecting technology acceptance [18]. According to Kamal et al. (2020) [19], the TAM has become the primary framework frequently employed in research on technology and the Internet of Things (IoT). Furthermore, it is commonly used in studies investigating the interaction between technology and individual behaviours and intentions [19].

The TAM was developed in 1989 by Fred Davis and adopted from the Theory of Reasoned Action (TRA) [18]. In certain cases, the TAM provides more accurate measures than the TRA. Malatji et al. (2020) [20] state that perceived ease of use (PEOU) and perceived usefulness (PU), two TAM metrics, might potentially replace the majority of technological acceptance measurements in the TRA. The perceived ease of use (PEOU) examines the system’s functionality and usefulness [12,20]. It represents the user’s perception of how easy it is for them to interact with the technology. On the other hand, perceived usefulness (PU) examines the system’s convenience, which is how much users feel the system improves their job performance or effectiveness [12,20]. Together, these two TAM measurements provide insight into users’ views and intentions regarding adopting new technologies.

As time passed, the scope of the TAM widened, and its measuring factors improved, resulting in more dependable and precise results. Because of these advancements, the TAM is commonly employed by researchers conducting healthcare-related studies. This is due to the TAM’s substantial dependability, high test–retest reliability, and straightforward approach, which simplifies complex technology acceptance assessments by focusing on two key components.

#### 2.1.2. The Unified Theory of Acceptance and Use of Technology (UTAUT)

Information systems (ISs) and information technology (IT) have seen significant development, and this has attracted the attention of both researchers and practitioners, encouraging them to embrace and use them. Several theoretical models, including the TAM and TRA, have been used to study how ISs and IT have been adopted and accepted in recent decades. Nevertheless, these theoretical frameworks were first limited to Technological Acceptance Theory and did not incorporate the technological experience component. For this reason, among others, Venkatesh et al. (2003) [21] created the Unified Theory of Adoption and Utilisation of Technology (UTAUT). Since it was developed, it has been one of the most frequently used models to comprehend the adoption and utilisation of ISs and IT [5]. The original UTAUT model accurately represented a significant portion of the variation in behavioural intention and usage behaviour.

The UTAUT model has four key constructs, which are performance expectancy, effort expectancy, social influence, and facilitating conditions. Applying these four key constructs streamlines the assessment of technology acceptance. These constructs are essential determinants of use behaviour or intention to use, acting as critical indicators of technology acceptance.

Performance expectancy (PE) pertains to how much a person perceives that utilising the system will improve their job performance [22].Effort expectancy (EE) denotes the perceived simplicity of using the system [22].Social influence (SI) indicates the extent to which an individual believes that significant others expect them to adopt the new system [22].Facilitating conditions (FCs) are the degree to which a person believes that technical and organisational resources are available to facilitate the use of the system [22].

Together, these factors provide a comprehensive base of knowledge of the factors affecting people’s acceptance and utilisation of technology.

The UTAUT model thoroughly describes how variables affecting behaviour and intention change over time. According to Venkatesh et al. (2003) [21], the UTAUT may explain as much as 70% of the variation in intention. However, its ability to demonstrate specific acceptance and usage decisions within an organisation may be limited [14]. The UTAUT includes four moderators for dynamic elements, including user experience, organisational environment, and demography. This consolidates the current theoretical viewpoints and enhances research on individual acceptance. These improvements have made the UTAUT one of the most popular theoretical frameworks in IT/IS research [23].

### 2.2. Applications of the TAM Theory in Healthcare Industry Research

Technology has changed several industries in this digital age, such as manufacturing and transportation. In contrast to the transportation and industrial industries, the healthcare industry has been uneasy about adopting these technological innovations. Figure 1 depicts the Technology Acceptance Model (TAM), which has been used in several studies to explore the slower adoption rate of technology. These studies provide insights into the barriers to technology adoption in the healthcare field and propose viable solutions.

According to Zoccarato et al. (2024) [12], Italians have experienced difficulty in utilising a blood sugar monitor device called Dexcom ONE for various reasons, including functional, psychological, and rational factors. As diabetes, as well as its multifaceted complications, is always one of the most serious problems that occurs on a global scale, this device that can monitor blood sugar levels on a daily basis, as well as offer technological solutions for daily diabetes management, was developed for those who struggle with blood sugar control. Nevertheless, despite all of the advantages it offers to the intended audiences, it continues to face challenges in gaining societal acceptance, such as the technological design of the equipment as well as the psychological aspects of the patients. Consequently, the TAM theory was used in this study to identify the variables influencing the adoption of new technologies. The authors incorporated several components into the model to provide a reliable result. In addition to the core TAM theory variables of perceived usefulness and perceived ease of use, they included subjective criteria, as well as other factors such as glucose data visibility, trend arrows, alarms, stigma, and trust. In addition, they included control factors such as gender, age, the presence of carers, the kind of diabetes, and health literacy in the model. In order to determine the impact of psychological aspects, functional components, and rational constructions on technology adoption, they employed questionnaires to gather data from 157 respondents who had been using Dexcom ONE for at least one month. After examining the data with the SPSS system, they observed a significant correlation between rational thinking and the desire to use CGM technology. This discovery emphasises that doctors play an essential role in promoting the use of CGM technology. Not only do doctors play a crucial role in ensuring that CGM devices are efficiently used and accepted by diabetics, but so do manufacturers, patient groups, and healthcare providers.

Ekaimi et al. (2024) [5] studied the factors that influenced the use of teleconsultation during the COVID-19 pandemic. Because of the virus’s fast spread, teleconsultation technology was adopted more swiftly. As a result of the development of teleconsultation, which aims to limit the likelihood of face-to-face meetings, it was thought that it may be possible to prevent the COVID-19 virus from spreading by using teleconsultation rather than organising in-person sessions on a regular basis. However, many parties faced significant challenges as a result of the fast growth of teleconsultation along with the spread of the pandemic, including governments juggling legal issues, healthcare providers lacking the necessary equipment and training, and patients being leery of virtual therapy taking the place of in-person appointments. Therefore, the goal of this study was to examine patient behaviours associated with the adoption and use of teleconsultation during the pandemic, with a focus on identifying the factors that affected the adoption. In order to guarantee an accurate outcome, they included two more variables—trust and privacy—in addition to the TAM hypothesis in their analysis. To gather data, they developed an on-site questionnaire that was completed by 100 patients who were between the ages of 20 and 40, had a bachelor’s degree in education, and had used teleconsultation services in one of Indonesia’s private hospitals during the pandemic in 2020 and 2021. After the data analysis, they found a positive correlation between intention to use and perceived privacy, trust, and ease of use. Furthermore, there was a favourable correlation between intention to use and actual use. These findings led them to the conclusion that while older patients (those 55 and older) need additional support and instruction, patients, in general, find the programme straightforward to use. Patients have trust in this technology as their trust is built on previous contacts with the hospital as well as their awareness that they are interacting with licensed and trained medical personnel. Although patients with sensitive conditions continue to be concerned about data security, they are generally comfortable using the application because their medical records and consultation findings are kept up to date. Finally, patients are more likely to use the app if they find it convenient, safe, and trustworthy during online consultations.

Zin et al. (2023) [24] studied the use of digital healthcare tools by elderly Koreans. In light of Korea’s significant ageing population, older adults must learn about the newest medical technologies to enhance their safety. For example, they may prevent accidents by wearing smart health watches, which will also make it easier to monitor the health of older people. There has been a decline in in-person meetings due to the COVID-19 pandemic, as the virus spreads through these kinds of interactions. Because technology may significantly reduce the risk of viral propagation, its usage in the healthcare industry has increased tremendously as a result. However, physical constraints and a lack of technical understanding cause older adults to struggle with utilising technology. As a result, it is vital to understand their perspectives, attitudes towards evolving technologies, and willingness to use the newest medical technologies. Determining the barriers to digital healthcare technology adoption is also essential. As a result, they undertook research to evaluate the development and usage of digital health and wrist-worn wearable devices among senior Koreans using an expanded Technology Acceptance Model, taking into account the COVID-19 pandemic. Although they believe the original TAM theory had limitations for this kind of study, they nevertheless employed it in their inquiry. Thus, in order to account for different aspects, they added two more variables to the theory: social impact and facilitating conditions. They developed a questionnaire for the target population, which consists of Busan residents 56 years of age and older, in order to gather data. A total of 170 responses were collected to examine the data. The analysis of the data revealed that perceived ease of use, perceived usefulness, and facilitating conditions all had a favourable influence on views on the adoption of digital health wearables. However, social influence had little effect on perceptions of these technologies. Their conclusion, which lends support to the larger TAM and is in line with previous study findings, is that a positive perspective is highly connected with the behavioural intention to utilise digital health wearables, implying that the more positively they perceive these technologies, the more likely they are to use them.

Alhashmi et al. (2019) [25] performed research on the use of artificial intelligence (AI) in the UAE healthcare business. The UAE is aggressively emphasising AI integration to improve its government healthcare services. To ensure the effective use of healthcare technology, everyone involved, including patients, medical professionals, and others, must have a high level of acceptance of AI. This study used the ETAM theory, which is an extension of the TAM theory with extra variables to adapt to the research context. In addition to the original TAM constructs of perceived ease of use, perceived usefulness, attitude towards use, and behavioural intention to use, they added four additional variables to the TAM theory: managerial factors, organisational factors, strategic factors, and IT infrastructure factors. These combinations of variables make up the ETAM theory. They argue that the ETAM’s inclusion of these additional variables allows for the identification of critical success factors (CSFs) related to the adoption of AI in the healthcare industry. The purpose of this study was to assess how well the ETAM predicts the successful adoption of AI in healthcare. They devised a questionnaire to collect data, and 53 responses, including those from both IT and medical staff, were received from 13 health facilities in Dubai. After analysing the data, they observed that all of the variables were supported, with the exception of the strategic considerations, which had a negative association with perceived ease of use. This means that healthcare project managers have to consider these external challenges when improving TAM structures for AI implementation.

Nazari-Shirkouhi et al. (2023) [26] investigated the Technology Acceptance Model to improve user acceptability of e-services in healthcare systems. The spread of the worldwide epidemic, COVID-19, has altered the way people live. It has accelerated developments in both physical and digital services, as well as the adoption and acceptance of these technologies, particularly in healthcare systems. However, the acceptability of technology in the healthcare business is still being analysed. Therefore, a thorough model is required and would be advantageous to create effective communication in electronic healthcare systems. Several studies have proven that the TAM theory framework is reliable for predicting technology adoption in practically all industries, including the healthcare industry. This study proposed a TAM-based theoretical framework tailored to the healthcare industry in order to increase e-service uptake. In this study, several external variables, such as computer literacy, website quality, service quality, user satisfaction, and user attitude, were incorporated to allow for more comprehensive data collection and analysis of e-service usage. For data analysis, 216 responses to a series of questions that were created for the participants, which were from patients who had utilised the hospitals’ electronic services throughout the previous year, were gathered. Upon conducting an analysis of the collected data, they discovered that every hypothesis had a positive impact, with the exception of user attitude, which did not significantly affect the desire to utilise e-services inside the healthcare system.

Dhagarra et al. (2020) [13] investigated the effect of trust and privacy issues on the acceptance of technology in healthcare from an Indian viewpoint. Across the world, governments had decided to increase access to basic healthcare by the year 2000. Unfortunately, it is difficult for the destitute in developing countries like India to achieve these aims due to the expensive costs. Because of this, it is believed that the goal could be achieved by adopting technology in healthcare service delivery. However, both physicians and patients must embrace and use technology to adopt it successfully. Several research studies have examined the impact of technology on various aspects of healthcare, such as cost, efficiency, and quality. Nevertheless, Dhagarra et al. (2020) [13] found that the majority of research has concentrated on the standpoint of the service provider, and there are fewer studies focused on understanding patients’ perceptions of technology usage and how it relates to behavioural factors. Instead of examining the relationship between different behavioural components and the adoption of new technology in the healthcare industry, those studies concentrated on cognitive presumptions. Thus, Dhagarra et al. (2020) [13] performed this study to better understand the relationships that predict patients’ adoption of technology in healthcare services using the revised TAM theory, adding the variables of trust and privacy concerns, as they believe that these factors are critical for building and maintaining trust and ensuring trust for patients’ continued participation in a healthcare delivery system. To ensure that survey participants understood the questions, they prepared a narrative and asked them to read it before responding. A total of 416 responses were then collected and analysed. The data analysis found that trust positively correlates with patients’ intent to utilise healthcare technology, but privacy concerns negatively correlate with perceived ease of use, meaning that trust encourages technology adoption in healthcare but privacy issues do not. This indicated that addressing privacy concerns is crucial to increasing patient acceptance of new technology.

Walle et al. (2023) [27] conducted a study that employed a revised Technology Acceptance Model to forecast how healthcare professionals in resource-constrained situations would adopt electronic personal health record systems. With a focus on their intent to embrace such technology, this study aimed to evaluate healthcare professionals’ acceptance of electronic personal health record (ePHR) systems in resource-constrained settings. This is due to Walle et al. (2023) [27] discovering that despite the potential advantages—which include better patient–provider relationships, better medication adherence, improved health outcomes, and more—ePHRs face implementation challenges, such as a lack of awareness and reluctance among healthcare professionals. Despite being created with consumers in mind, PHRs are only as effective as the healthcare professionals who utilise and accept them. Consequently, it is imperative to devise strategies to enhance the integration of technology among healthcare professionals. Therefore, this study examines the acceptability of ePHRs by healthcare professionals by using a revised Technology Acceptance Model (TAM) to identify key contributing elements. In addition to the TAM variables of perceived usefulness and perceived ease of use, they included two additional variables—information technology experience and digital literacy—to measure health professionals’ behavioural intention to utilise ePHRs in low-resource settings. They developed a questionnaire, to which 638 medical professionals employed at Amhara regional state teaching hospitals responded. Two days of training were provided to supervisors and data collectors in order to ensure consistency and quality control. After analysing the data, they found that healthcare practitioners’ intentions to adopt electronic personal health records were significantly influenced by perceived usefulness, attitude, and perceived ease of use. Their attitude towards and level of digital literacy also had a significant impact on how willing they were to adopt these tools. These results underline how important it is to enhance usability, cultivate favourable attitudes, and raise digital literacy in order to boost Ethiopians’ acceptance of electronic personal health data.

Wang et al. (2023) [28] used the TAM to predict virtual reality’s acceptance among paediatric healthcare practitioners. The TAM theory is a well-known theoretical framework that has been utilised in a variety of research studies on technology acceptance, including perceived utility and ease of use. It has been approved for use and modified for use in numerous studies on the adoption of technology, including medical investigations. Virtual reality (VR) is an innovative technology with several uses that are starting to catch on in the market. Despite a paucity of research, people began to realise that VR could be useful in the healthcare industry, especially in the area of anxiolytic treatment for hospitalised children. Although widespread usage of VR in the healthcare industry is still restricted, several studies show that technology has potential in healthcare settings. Therefore, the purpose of this research was to evaluate the validity of the TAM in this particular healthcare setting and to model attributes that predict the behavioural intentions of paediatric healthcare professionals to use virtual reality as an anxiolytic for children with illnesses. To achieve a more accurate conclusion, Wang et al. (2023) [28] modified the TAM theory to include numerous characteristics such as age, gender, race, clinical role, years of experience, past VR usage, perceived enjoyment, intention to use, want to purchase, and curiosity. For the data collection, they prepared surveys for medical professionals and received 270 responses. They questioned non-healthcare clients in addition to the medical professionals. The data analysis revealed that perceived usefulness, pleasure, and perceived ease of use all had a significant influence on their willingness to use and purchase VR. Besides that, results obtained by non-healthcare consumers differ from those of healthcare professionals, in which the perceived ease of use was influenced by age, prior usage, price, willingness to pay, and curiosity. Based on these results, Wang et al. (2023) [28] concluded that VR may be employed in several applications based on the outcome. Moreover, the revised TAM may be used in the following studies on behavioural intentions.

Karkonasasi et al. (2023) [29] applied the Extended TAM theory in order to investigate the acceptability of a text-messaging vaccine recall as well as a reminder system in the healthcare sector in Malaysia. Despite the fact that, at present, technology can benefit other industries, the healthcare business still relies heavily on paper records for various issues, such as unfulfilled vaccinations, missed appointments, and other concerns. This results in a heavy workload for healthcare workers. Therefore, the goal of this study was to evaluate the efficacy of the text-message-alert-based Virtual Health Connect (VHC) vaccine recall and reminder system. The VHC approach was designed to automate patient recalls and reminders and, as a result, vaccination scheduling and completion while also streamlining the immunisation procedure for nurses. The TAM theory was applied in this study along with two additional variables, perceived compatibility and perceived privacy and security issues, in order to account for all the variables and provide a better result. Because the survey was conducted in English, the respondents were nurses with fluency in the language and a background in IT. Nurses were selected because they are the front-line employees in charge of vaccination data. Thus, their viewpoints on VHC are crucial to its effective implementation. Each nurse investigated the organisational acceptability of VHC, speaking on behalf of a government hospital or clinic. Following the questionnaire’s construction, it was put to content validity testing by two experts from Universiti Sains Malaysia School of Management, and the structural and linguistic adjustments were made in accordance with the specialists’ recommendations. To examine the reliability of the devised questionnaire, they conducted a pilot research study with 31 respondents, taking their feedback into account. However, these respondents were not invited to the final survey to verify the authenticity of the results. A total of 128 valid responses were collected for use in the data analysis. The findings revealed that perceived system compatibility had a significant impact on nurses’ attitudes, but perceived privacy and security issues had no discernible effect. Meanwhile, nurses’ attitudes were positively influenced by perceived usefulness and perceived ease of use. However, perceived usefulness had no noticeable influence on their willingness to utilise VHC, which was most likely owing to their lack of expertise with regard to the system. Based on these data, they concluded that nurses’ favourable opinions had a significant impact on their motivation to use VHC. In summary, perceived usefulness and perceived ease of use were deemed to be secondary variables, with perceived compatibility being the most significant. Healthcare decision-makers can benefit from results by increasing coverage of child vaccination and improving services. Researchers are able to aid in improving managers’ and software developers’ comprehension of system adoption in the healthcare industry, including VHC and related technologies. This study fills in the gaps in these systems’ acceptance features, which might lead to better ways of implementing healthcare innovations into practice.

### 2.3. Applications of the UTAUT Theory in Healthcare Industry Research

In addition to the TAM theory, IT-related research frequently employs the UTAUT theory. Figure 2 is provided as a graphic illustration of the UTAUT theory for easy understanding. In contrast to the TAM, the UTAUT is a theory made up of eight different independent acceptance theories [21,30]. By merging these theories into one model, the UTAUT streamlines complex technology acceptance assessments and offers a comprehensive knowledge of technology adoption in the healthcare industry [21]. It aids in understanding the elements that influence technology adoption and utilisation, either positively or negatively. This study assessed various papers that employed the UTAUT theory to gain a better understanding of how it supports research on technology adoption in the healthcare industry.

Nurhayati et al. (2019) [31] investigated the UTAUT model’s ability to predict health information system adoption. With today’s technological advancements, information technology (IT) has greatly benefited society. Despite understanding that IT integration could improve quality and reduce costs, the healthcare sector encountered significant obstacles in embracing IT. Numerous studies demonstrated that without the use of IT, healthcare administration risks becoming inefficient and losing patients’ trust. The digital tools associated with IT adoption include those for managing chronic diseases, promoting health, preventing illness, diagnosing ailments, and improving accessibility to healthcare, in addition to reducing medical errors and costs. Therefore, it is essential to promote IT adoption in the healthcare sector. However, the healthcare business frequently experiences delays in deploying IT solutions due to the difficulties of IT adoption. This study sought to determine the aspects that primary healthcare nutrition officers considered while implementing nutrition information systems. To this end, this study made use of the UTAUT theory, which offered a reliable framework for forecasting user acceptance of IT in the healthcare industry. Additionally, this theory could also provide insightful guidance on how to overcome adoption obstacles and guarantee effective implementation. In order to analyse the data, 50 questionnaires were completed by the officers who used the nutrition information system as participants. After analysing the data, they concluded that behavioural intention, performance expectation, effort expectation, and societal effect are essential factors to consider when putting a health information system into place.

Osifeko et al. (2019) [22] employed a modified version of the UTAUT theory to study e-health services. E-health is one of the types of modern technology that is being used in the healthcare industry. It aims to make healthcare delivery more efficient while enabling experts and individuals to perform activities that were previously impossible. Adopters benefit from cost-effective care delivery, accurate diagnosis, quicker access to a patient’s medical history, and other advantages. However, the high cost of this technology make it impossible for many to afford, especially in low- and middle-income nations like Nigeria. Consequently, many involved in the healthcare industry are searching for innovative approaches to remove financial barriers to the provision of healthcare services. The adoption of this technology is one of the elements limiting the growth of e-healthcare, in addition to its high cost. Hence, the goal of this study is to evaluate the acceptability and utilisation of e-health services in Lagos, Nigeria, by implementing a modified version of the UTAUT theory, which incorporates the variables identified in prior research as well as focus group discussions such as attitude, computer phobia, self-efficacy, ICT infrastructure, e-health policy, and e-health knowledge. A survey instrument, which was a questionnaire, was developed in order to gather data from people who utilised e-health services at ten different medical facilities. A total of 210 responses were compiled and assessed. Through the analysis of the responses, the authors found that every variable affected how widely the system was adopted. Nonetheless, it was discovered that social influence had the most significant role in Nigeria’s adoption of e-health. Therefore, they concluded that users were more likely to use the system if they received encouragement to do so from superiors or coworkers.

Zhou et al. (2019) [30] assessed the social effect and enabling variables that encourage Ghanaian nurses to utilise hospital electronic information management systems (HEIMSs) by using the UTAUT model. HEIMSs are among the health information technologies (HITs) that healthcare institutions throughout the globe have allocated resources towards. However, it was found that the adoption of HITs was hindered by the nurses who used it, who had low levels of technical acceptance. Consequently, the goal of this study was to evaluate the social influence of nurses and their impact on HEIMS-using behaviour, as nurses are the largest group of hospital employees and the cornerstone of healthcare delivery. They thus have a significant effect on the implementation and assessment of HEIMSs in Ghana. In addition, the goal of this study was also to examine the facilitating conditions in hospitals and the behavioural intentions of nurses that influence the habit of utilising HEIMSs. Thus, the UTAUT theory was used in this study, retaining the facilitating conditions (FCs) and social influence (SI) factors as assessment elements but removing the performance and effort expectations variables. They developed an electronic platform questionnaire and sent it to nurses working at five major public hospitals in Ghana that were using HEIMSs for patient care in order to collect data. A total of 660 replies were collected for the data analysis. Based on the data analysis, they found that social influence and behavioural intentions had a significant impact on nurses’ adoption of HEIMSs and their use behaviour. Moreover, the behavioural intentions of nurses in Ghanaian hospitals to use HEIMSs were most influenced by facilitating conditions and social influence. Based on their research, they concluded that hospital managers should support and encourage nurses to accept and use HEIMSs while simultaneously fostering a positive work environment for them to do so. In addition, hospital administration needs to provide nurses with the right IT resources so they can utilise and comprehend the HEIMS. In conclusion, this study provides an empirical and scientific foundation for better understanding the behavioural issues related to the adoption of HEIMSs by hospital nurses in Ghana. Besides that, the findings may also contribute to the existing body of knowledge, raise awareness, and encourage further research in developing nations.

Zhu et al. (2023) [14] studied the use intention of mHealth apps in China using the UTAUT-2 theory. Due to the outbreak of the COVID-19 pandemic, the shortcomings of traditional healthcare, such as higher wait times, a scarcity of medical and nursing staff, and so on, were highlighted. Fortunately, with the fast growth of the internet and technology, an application called mobile healthcare (mHealth) was created to address recurring issues in the healthcare system, such as a lack of resources and conflict between physicians and patients. However, there are other concerns with current mHealth apps, including a lack of innovation and standardisation. Therefore, this study was carried out to address the limitations of the present mHealth applications by concentrating on user demands and offering improved product design recommendations. Furthermore, this study also aimed to identify the critical factors that influence user intention to use mHealth applications in order to promote their usability, satisfaction, and broader adoption in healthcare settings by utilising the UTAUT-2 theory to guide the design and development of mHealth technologies. In their research, price value, perceived risk, and perceived trust are the two additional factors that have been introduced to the UTAUT-2 hypothesis. In addition, they increased the number of components associated with each variable to account for all potential outcomes, including time costs and media influence, among other things. In order to gather information, they created a questionnaire, tested it with thirty respondents who were college students and psychological specialists to make sure all the questions were understood, and made adjustments to the surveys in response to their input. A professional platform was used to send the surveys, and 371 legitimate responses were obtained in total. After analysing the data using SPSS and AMOS, they found that aside from price value, performance expectancy, effort expectancy, social influence, facilitating conditions, and perceived trustworthiness all had a significant impact on user intention. Meanwhile, the result of this study has indicated that the perceived risk has a negative influence on user intention, which suggests the need to resolve security and privacy concerns. Therefore, this study suggested raising emotional engagement and addressing risk perceptions in order to enhance user experience and motivate users to utilise mHealth applications. Ultimately, this study’s conclusions provided a theoretical basis for the functional and user interface design of mHealth apps in the future, as well as for more aggressive marketing of this technology.

Philippi et al. (2021) [32] conducted a study on the acceptability of digital health interventions to validate and develop the UTAUT theory. The growth of technology has supplemented and enhanced the healthcare business, for example, through internet- and mobile-based interventions (IMIs). IMIs have the ability to go around structural obstacles due to their adaptability to changing times and locations. Additionally, they provide a low-barrier-to-entry therapy option that might help lessen or perhaps completely eradicate feelings of shame, prejudice, and so on. Although IMIs have been researched and found to be helpful for a variety of medical conditions, including both physical and emotional, patients and medical professionals have a low-to-moderate level of acceptability for them, which underscores the need to understand the variables influencing IMI uptake since they are likely contributing to the low rates of adoption and adherence. Thus, the purpose of this study is to further validate the UTAUT model in relation to IMIs. The objective is to comprehend acceptance predictors and discover ways to improve acceptance of IMIs, such as by including internet anxiety in the UTAUT theory, as was the case in this investigation. To conduct the data analysis, they used the keywords “mobile”, “internet”, “online”, “smart”, “web”, “blended”, “acceptance”, and “intention” to search for relevant research papers that would provide the original data for this study to conduct the secondary analysis. Following the initial search, two different researchers further checked the complete texts, abstracts, and titles of the remaining studies for eligibility. Accordingly, this study referenced a total of 14 papers. Those studies that were able to provide original data were also included in this study. In total, 10 contributed primary data. Following data analysis, this study’s conclusions demonstrated that performance expectancy, effort expectancy, and social influence were all relevant to IMI. On the other hand, the acceptance of IMI was not significantly impacted by the traditional moderating variables. However, it was shown that effort expectation and social influence, as moderator, were considerably affected by the recently included variable, online anxiety. This indicates that social support is vital for those who are more anxious while using the internet, and that lowering anxiety and increasing acceptance may be achieved by providing enough information, ensuring data security, and offering technical assistance. In addition, performance expectancy has been shown to have a significant impact on acceptance, indicating that this should be the central area of effort for boosting IMI’s efficacy and acceptability. In conclusion, it was adequate to apply the UTAUT theory to the setting of acceptance of IMI.

Farhady et al. (2020) [33] performed a study to assess the variables affecting the adoption of mobile health technology by applying the UTAUT theory to the situation of difficulties arising from blood transfusions in patients with thalassemia. The advancement of technology has led to the development of an application named mobile health, also known as mHealth, which removes time and location restrictions to provide healthcare services to everyone, anytime, as the quality and coverage of healthcare indices rise. Despite the fact that computers can significantly improve treatment quality in terms of paperwork reduction and convenient access to patient data, other research on the adoption of technology in the healthcare industry, including electronic health records (EHRs), revealed that more conservative and older doctors were less likely to use computers in the treatment process. Aside from EHRs, the mHealth application is among the technologies that have struggled with doctors’ and nurses’ lack of adoption in the healthcare industry. Consequently, Farhady et al. (2020) [33] carried out this study with the goal of identifying the factors that affect the adoption and acceptance of technology, particularly mobile health technology, among haematologists, using the Unified Theory of Acceptance and UTAUT. This is because mHealth has the potential to significantly enhance the healthcare sector; thus, it is essential to increase its acceptability. For example, it could be used to assist patients with thalassemia in minimising the negative effects of blood transfusions. Farhady et al. (2020) [33] designed a modified UTAUT theory based on the revised UTAUT theory that was adapted from Schaper and Pervan (2007) [34], placing a greater emphasis on social and technological difficulties. They created a questionnaire and disseminated it to 58 specialist haematologists in order to collect the data for data analysis. Through data analysis with SMART PLS2 and SPSS, they discovered that the greater accessibility and dependability of this technology among haematologists would result in a decrease in the adverse consequences of blood transfusions for patients suffering from thalassemia. Additionally, they discovered that this sector had not been investigated previously, so they recommended carrying out additional research because many of the technology’s principles require greater scholarly attention, particularly in Iran.

Yin et al. (2022) [35] used a modified version of the UTAUT theory to investigate the user acceptability of wearable intelligent medical devices. Precision medicine is the driving force behind the growing interest in intelligent medical goods, for example, wearable intelligent medical devices (WIMDs). These artificially intelligent gadgets record physiological indicators and track metabolic states to enable users to continually monitor their fitness and health. Current commercially available WIMDs essentially monitor vital signs but they face hurdles, including privacy issues and lack of a user-friendly design, despite their potential for illness diagnosis and therapy assistance. Therefore, the study of variables impacting acceptability and behavioural intention to use WIMDs is essential in order to improve WIMD usability and increase their role in medical practice. Thus, this study aimed to further this knowledge by examining these characteristics and their effects on the use of WIMDs. To better address all the characteristics of this study, the researchers integrated the theories of TPR, TAM, and UTAUT to design a modified UTAUT theory. In order to collect data for analysis, they created a questionnaire, circulated it using the WeChat app, and successfully gathered 2192 valid responses. By using SPSS for analysis, they found that the idea that perceived cost had a negative impact on the intention to use WIMDs was not supported by their data, while all other hypotheses were validated. To be more precise, perceived cost and behavioural intention were positively correlated, but health expectancies, perceived ease of use, and social influence all had a favourable impact on behavioural intention, which in turn affected WIMD usage habits. They concluded that the adoption of WIMDs is driven by enabling factors such as robust support networks and efficient after-sales services. Adoption intentions are positively impacted by health expectations; however, expectations are lower for those with underlying health concerns.

Yousef et al. (2021) [36] investigated patients’ intentions to use a personal health record by applying the revised UTAUT theory and conducting a secondary data analysis. Global healthcare delivery has changed since the early 21st century, which led to people being increasingly urged to take greater responsibility for their health through e-health tools like personal health records (PHRs) due to the growth in chronic illnesses and developments in information and communication technology. PHRs are a type of e-health tool that improves patient empowerment and participation by allowing users to track and manage their health information online. These records enable organisations to provide the best possible healthcare services to patients as healthcare companies may enhance treatment quality, lower costs, and expand access by utilising PHRs to store health information. The Ministry of National Guard Health Affairs (MNG-HA) introduced its PHR system in 2018, which was termed MNGHA Care. It has functions that include monitoring test results, making appointments, seeking reports and medication refills, and reminding people to be immunised. In addition, patients have the ability to self-evaluate their discomfort, performance status, and quality of life by uploading their personal health data. Despite the initial high level of interest, multiple studies on patient uptake after implementation are necessary. Thus, the purpose of this study was to apply the UTAUT theory to discover parameters that predicted patients’ desire to use the MNGHA Care PHR. Based on the critical evaluation and the setting of the study, they deleted the voluntariness of usage factor because the study’s participants were volunteers, included attitude as a variable, and used health state as a moderator. The data analysis for this study was secondary data analysis; the data used in this study were primarily gathered by Hoogenbosch et al. [37] for their research between December 2019 and February 2020. Moreover, they also added minor modifications and additional items according to the objectives of the study. A total of 546 persons filled out the survey in the initial research. However, only 324 participants provided answers to all questions related to the MNGHA Care PHR in the secondary data analysis. By analysing using SPSS, performance expectancy, effort expectancy, and a positive attitude were found to strongly predict behavioural intention, underscoring the significance of attitude in PHR uptake. Social influence had a detrimental impact on patients who had experience using health apps, while it enhanced behavioural intention in those who did not. The assumption that attitude is a key predictor of technology usage is supported by the result that a favourable attitude towards PHRs significantly impacts behavioural intention. This finding also shows the positive attitudes among peers and healthcare professionals towards promoting the adoption of PHRs.

Wrzosek et al. (2020) [38] conducted a study to explore doctors’ perceptions of e-prescribing following its mandatory adoption in Poland, utilising the UTAUT theory. With the launch of e-health initiatives in 2008, e-prescribing became a crucial component of Poland’s healthcare system. It made healthcare procedures more automated. E-prescribing became necessary nationally on 8 January 2020, with the exception of a few particular cases, such as prescriptions for authorised imported medications or cases where there were problems with system access. Research has shown that e-prescribing can improve healthcare quality, save expenses, and minimise mistakes. Despite these advantages, embracing new technologies is frequently tricky for healthcare workers. For their implementation to be successful, it is critical to achieve a deeper understanding of the elements impacting their acceptability. Therefore, this study employed the UTAUT theory in order to determine these parameters and encourage the use of electronic prescription systems. For the purpose of gathering data, they created a questionnaire, which they then sent to primary care physicians in Poland’s private sector. For the purpose of the data analysis, 381 valid replies in total were gathered. After analysing the data, this study indicated that age stereotypes have not materially impacted doctors’ embrace of technology, although numerous studies have shown that seniority frequently affects acceptance, with more senior physicians generally being less persuaded of its advantages than their junior counterparts. However, it emphasises how crucial customised e-tool designs and standardised software are to raising acceptance rates. The study concluded by suggesting that end users’ expertise levels and a focus on user-friendly interfaces are critical factors to take into account when creating health system e-tools. Besides that, providing efficient training could improve adoption, as it makes the advantages of new technologies abundantly evident. The study also discovered that elements critical to promoting and maintaining the adoption of new technologies included providing sufficient financial and technical assistance throughout implementation.

Christian et al. (2023) [39] conducted a study to examine the factors that influence the use of e-healthcare among Generations Y and Z during the third year of the COVID-19 pandemic by employing the UTAUT theory. People are very aware of the need to take personal protective measures, despite the number of cases decreasing in Indonesia during the third year of the COVID-19 epidemic. Simultaneously, e-health, or mobile health, services are now much more accessible, and service providers are actively working to improve consumer acceptability. With the help of models like the TAM and UTAUT, this growth has sparked ongoing conceptual discussions in research that have primarily focused on variables like trust risk, self-efficacy, effort expectations, performance expectations, facilitation conditions, and social influence as essential factors influencing the adoption of health apps. However, the significance of participant age has frequently been disregarded in these investigations; thus, this study was performed using a modified UTAUT theory, which added several variables, including attitudes towards using technology, perceived ease of use, and perceived usefulness, to cover all the bases and produce more accurate results in order to close this gap. Therefore, this study was conducted to investigate the specific factors shaping attitudes and behaviours towards e-health services among Generations Y and Z. For the data collection, they created a questionnaire and gave it to the participants. They were able to obtain 268 valid replies, of which 134 came from Generation Y and 134 from Generation Z. According to this study, neither generation showed a significant relationship between their behavioural intentions towards utilising technology and performance expectancy, effort expectancy, social influence, or enabling factors. In addition, this study discovered that Generation Z demonstrated a significant relationship between behavioural intention and usage behaviour, indicating a more flexible posture towards technology. Meanwhile, Generation Y’s attitude towards technology was not affected by behavioural intention. Furthermore, Generation Z showed that perceived ease of use strongly impacts their behavioural intention, whereas Generation Y showed no significant relationship with perceived ease of use. Nonetheless, perceptions of usefulness have a substantial impact on behavioural intention, which in turn affects usage behaviour, according to both generations. As a result, these data demonstrate clear generational differences in the adoption and use of technology, with Generation Z showing a more adaptable and tech-savvy attitude towards embracing healthcare innovations than Generation Y.

### 2.4. Critical Evaluation of the TAM and UTAUT in Healthcare Contexts

The theoretical frameworks used in these studies, the TAM theory or UTAUT theory, reflect various viewpoints, methods, and findings. However, the goal of every study is to promote societal acceptance of technical advancements in the healthcare industry. However, summarising these articles alone can leave out essential details. As demonstrated by Zoccarato et al. (2024) [12] and Ekaimi et al. (2024) [5], the TAM and UTAUT are fundamental models. However, their straightforward application often overlooks critical external factors, including cultural differences, trust dynamics, and domain-specific regulatory issues. These gaps highlight the necessity for additional context-sensitive modifications.

The effects of perceived ease of use or performance expectations are two examples of predicted associations that are often confirmed by TAM and UTAUT studies. Despite this, they rarely examine how these factors interact in complex real-world healthcare settings. For example, the research carried out by Zin et al. (2023) [24] demonstrated that stigma and computer literacy have an impact on the adoption of healthcare tools among older adults. However, the study did not discuss how social support networks can help reduce these barriers. In addition, Walle et al. (2023) [27] also emphasised the importance of digital literacy for healthcare workers. Nevertheless, they failed to examine organisational actions to bridge this gap. These omissions limit the practical use of the findings and highlight the need for a more comprehensive synthesis of the literature.

Together, these results demonstrate that the TAM’s simplicity often leads it to overlook complex obstacles to healthcare adoption. However, as noted by Zhou et al. (2019) [30], the integrative architecture of the UTAUT may be too complicated and resource-intensive for real-world implementations despite its theoretical strength. These findings indicated that an integrated strategy is necessary to integrate their advantages while addressing the shortcomings of both paradigms.

Identifying gaps can also reveal the direction of future research. For example, although Ekaimi et al. (2024) [5] emphasised trust as a crucial factor in the adoption of teleconsultation, the role of trust-building interventions remains underexplored. Similarly, Zhu et al. (2023) [14] also discovered that perceived trustworthiness was a factor in the adoption of mHealth. Still, they failed to consider how to develop user-centred apps that foster trust. Therefore, exploring these areas could provide actionable strategies to address resistance to healthcare technology adoption.

These significant findings will serve as the foundation for this study’s analysis of how the TAM and UTAUT could be modified to meet specific healthcare industry concerns. In order to ensure that theoretical models are both practically significant and contextually relevant, this study attempts to establish a strong basis for future research by synthesising current findings and identifying actionable gaps.

## 3. Materials and Methods

### 3.1. Research Methodology

To ensure the quality of the study, a number of stringent criteria were applied during the narrative review process. First of all, the core goal of the review was to screen existing studies related to the application of the Unified Theory of Acceptance and Use of Technology (UTAUT) and the Technology Acceptance Model (TAM) in the healthcare industry. The studies that were included were carefully chosen to ensure that they included a range of TAM and UTAUT applications in healthcare and incorporated a diversity of healthcare technologies, locales, and user demographics. This approach encompasses the complex barriers and flexibility required for the industry to adopt technology. A total of 20 papers that were closely connected to the application of the TAM and UTAUT in the healthcare industry were added following the completion of pertinent exclusion and screening procedures.

Throughout the screening procedure, this study employs keywords such as “TAM”, “UTAUT”, and “healthcare” to perform a thorough search to find highly relevant research papers that addressed such issues. At the same time, this study only included publications published between 2019 and 2024 in order to guarantee the timeliness of data and viewpoints and to make sure the review represents the most recent research findings. This is due to the concern of quick development in healthcare technology along with the difficulties that come with it; only research studies that are published within a specific range of time are involved in order to maintain the consistency of this study. By focusing on recent years, this research aims to capture the advancements and most recent trends that are most relevant to contemporary practice. This analysis excludes the underlying research conducted before 2019 since it has already been thoroughly assessed in earlier evaluations. Instead, it seeks to address current development challenges while expanding upon that core work. This approach ensures that the findings remain up to date and consistent with the most recent research in the field.

Last but not least, the chosen studies are found from reputable and authoritative sources, such as publications from the *British Medical Journal* (*BMJ*), ScienceDirect, SpringerLink, *GBFR*, *International Journal of Information Technology and Language Studies* (*IJITLS*), *JAMIA Open*, MDPI, *Kemas*, *JMIR*, and *Journal of Experimental Research* (*JER*). This is to guarantee that the included articles have passed stringent quality control and peer-review processes, which is essential in offering superior research data and insights.

Figure 3 displays the PRISMA flow chart for this investigation, which describes the precise screening procedure and the inclusion of pertinent papers.

First, a total of 141 records were gathered from various trustworthy sources, including the *BMJ* (n = 8), Sage Journals (n = 14), ScienceDirect (n = 20), SpringerLink (n = 16), Research Gate (n = 12), Beadle Scholar (n = 5), *GBFR* (n = 9), *International Journal of Information Technology and Language Studies* (*IJITLS*) (n = 6), *JAMIA Open* (n = 5), MDPI (n = 17), *Kemas* (n = 3), *Journal of Experimental Research* (*JER*) (n = 2), *JMIR* (n = 19), and *Procedia of Social Science and Humanities* (*PSSH*) (n = 5).

Following the inclusion of these records, a total of 9 duplicate records were eliminated through screening, leaving 132 records for the second review phase. Out of these records, 72 entries were removed because the use of TAM or UTAUT frameworks to investigate medical technology adoption was not demonstrated in their titles or abstracts. Following the evaluation of the remaining 60 records for eligibility, 48 records were disqualified for a variety of reasons: 18 of them were not published between 2019 and 2024, 16 of them did not specifically address the TAM or UTAUT framework, and 14 of them did not deal with the healthcare industry.

Following the evaluation, only 12 research articles remained. Thus, 8 more studies that satisfied the inclusion criteria were added to the review process, increasing the total number of studies examined to 20. The results of this study are more trustworthy because of the stringent screening procedures that guaranteed the sample’s relevancy.

### 3.2. Meta-Analysis

#### Narrative Synthesis (Qualitative Meta-Analysis)

Traditional meta-analysis cannot be used in this study due to the lack of unambiguous quantitative results as well as the lack of effect estimates, statistics, or raw data in the included review studies. As a result, a narrative synthesis technique is employed in this study in order to explore psychological obstacles to medical technology adoption using the TAM and UTAUT frameworks. The main patterns and similarities, which may also be interpreted as recurrent themes, were found by combining quantitative discoveries like statistical correlations with qualitative insights like thematic analysis of case studies. These results were then compiled into a concise and well-organised synopsis that aims to completely describe the primary obstacles and enablers influencing the adoption of medical technology.

Both qualitative and quantitative research methods were used in the examined papers, which covered a wide range of sample sizes. For example, one of the studies included in this analysis, the study conducted by Nurhayati et al. (2019) [31], had the smallest sample size, with responses collected from 50 nutrition officers, using the UTAUT framework to anticipate their adoption of health information systems. On the other hand, Yin et al. (2022) [35] conducted the study with the largest sample size of those evaluated in this study, using an upgraded version of the UTAUT framework to investigate the acceptability of wearable smart medical devices among 2192 users. This broad range encompasses both small-scale qualitative enquiries and large-scale quantitative research, reflecting the diversity of study situations. Additionally, in order to obtain an accurate conclusion, some of the research included in this analysis reanalysed data from earlier investigations. For example, the study by Philippi et al. (2021) [32] leveraged existing data to explore the adoption of digital health therapeutics, broadening the research perspective and enhancing the validity of the findings.

By reviewing relevant studies, this study investigated the psychological elements that facilitate and impede the adoption of medical technology, based on the TAM and UTAUT frameworks. Although some studies were unable to provide specific data, many have demonstrated the importance of psychological factors, such as perceived ease of use, performance expectations, social influence, and effort expectations. For example, Zin et al. (2023) [24] emphasised the technological difficulties faced by the elderly when using digital health solutions, while Wang et al. (2023) [28] observed that the ease of use of virtual reality technology significantly influenced the adoption of this technology by paediatric medical professionals. On the other hand, according to Yousef et al. (2021) [36], performance expectations have a significant impact on the adoption of personal health records (PHRs). In contrast, Zhu et al. (2023) [14] found that users’ performance expectations of mobile medical applications strongly influenced their intention to use them. Additionally, Zhou et al. (2019) [30] further demonstrated how social influence has a significant impact on nurses’ behavioural intentions in environments with few resources. Osifeko et al. (2019) [22] assert that social influence is essential in motivating low-income countries to embrace e-health services. Furthermore, Nurhayati et al. (2019) [31] emphasised that effort expectations have a significant impact on the adoption of health information systems. These studies demonstrate the need to overcome organisational and psychological obstacles, as well as to increase managerial support and resource investment, in order to successfully deploy new technology in the healthcare industry.

By using the narrative synthesis analysis, this study enables us to identify the barriers that influence the adoption of healthcare technology despite the lack of a quantitative meta-analysis. The study’s findings demonstrated that one of the most important ways to advance the development of healthcare technology is to improve the perceived ease of use, performance expectations, social influence, and effort expectations to overcome psychological and organisational barriers.

## 4. Discussion

### 4.1. Literature Analysis

All of the included studies that were utilised in this review are stated in Table 1, along with their topics, participants, and sample sizes, which makes it easy to evaluate and have a thorough grasp of those studies.

On the other hand, Table 2 includes the theory they employed as well as the modifications they made to it to better fit into their research. The tick (/) is used to indicate the theory being applied in the related study. This table shows that every study that used the TAM theory had to make changes to the theory in order for the study to be conducted. This suggests that the TAM theory requires the addition of additional variables in order for it to be accurately applied in healthcare-related research, as its two key constructs are insufficient to investigate the factors influencing the adoption of technology in the healthcare industry.

Simultaneously, this table also shows that the UTAUT theory needs to be modified in order to be used in studies pertaining to healthcare. Nevertheless, two studies were carried out without modifications to the UTAUT theory, suggesting that the theory has a thorough framework that is applicable in healthcare settings even in the absence of changes. As there were only two studies that applied the original UTAUT theory, this result might be biased.

Furthermore, one study that was included in this review employed an integrated theory that included both the UTAUT and TAM theories in their research. Their research suggests that the integrated theory may have produced a reliable conclusion, but in order to account for all the potential factors, they still needed to incorporate more variables. However, as there was only one study that employed the integrated theory, this conclusion might be biased.

Table 3 provides a more comprehensive overview of the advantages and limitations of the TAM and UTAUT theories.

The TAM theory is a theory that several academics have extensively validated in a variety of contexts and industries. They have confirmed that it can provide a solid foundation to support empirical research. It is a generally accepted theory because it simplifies the difficult task of assessing technology acceptability so that both the participants and the researchers can perform the study with ease. It makes use of two key constructs, which are the perceived usefulness and perceived ease of use, to enable speedy implementation and analysis. Furthermore, it is also a theory that is highly flexible and applied in complex settings like healthcare settings by including other variables like perceived risk and perceived trust. However, although the TAM theory does have certain advantages, it also has a few limitations. First of all, it may overlook the factors that could affect the adoption of technology in a complex setting like the healthcare setting, as it simplifies the variables to only the perceived usefulness and perceived ease of use. Because of this, the researchers had to include more variables in order to obtain a precise conclusion or to make their study fit into a complex setting.

On the other hand, the UTAUT theory is a relatively new theory that has been effectively used in several study fields and has garnered substantial empirical support across a range of industries. This is due to the fact that it contains a comprehensive framework that enables direct or minor modification for application to a complex setting. Its four main components can offer an integrated perspective on the adoption of technology, which are performance expectancy, effort expectancy, social influence, and facilitating conditions. By utilising these constructs, the UTAUT theory may be applied to assess technology acceptance, resulting in a more accurate understanding of the various factors influencing technology adoption. However, it does have several limitations when applied. First, it is more difficult to use and comprehend than the TAM theory because it has a more comprehensive framework with more constructs. Moreover, it necessitates that the researchers have a solid grasp of the theory because of its complexity, and it also takes a long time to apply because it needs to be applied carefully. Additionally, it is less adaptable than the TAM theory in terms of adaptation and modification to particular settings without compromising theoretical integrity because it is an integrated theory that draws from eight other theories.

The UTAUT integrates the factors of eight acceptance theories to solve more general organisational and societal issues. At the same time, the TAM is based on psychology through the Theory of Reasoned Action (TRA) and focuses on human behaviour and intention. As a result, the TAM is especially well suited for individual adoption in the healthcare industry, such as when a patient or clinician interacts with a particular product. Nevertheless, the UTAUT is more appropriate for evaluating adoption at the organisational level, especially in environments with limited resources where infrastructural and social issues must be considered.

Though the simplicity of the TAM promotes wider diffusion, the empirical study shows that this simplification often requires an extension to deal with complex situations. Meanwhile, the UTAUT is comprehensive and could explain up to 70% of the variance in technology adoption behaviour, but it is not easy to deploy without a lot of expertise and data. In conclusion, although the UTAUT offers comprehensive insight into making it suitable to study complex adoption scenarios, the TAM stands out on the grounds of accessibility and flexibility. In turn, the choice between the TAM and UTAUT depends on the objectives of the study and the complexity of the environment studied.

### 4.2. Critical Analysis

As indicated by Table 2, a total of 20 existing studies that employed either the TAM, UTAUT, or both theories in their studies were included in this study in order to evaluate the differences, advantages, and disadvantages between the UTAUT theory and the TAM theory in the healthcare industry. These studies were investigated from numerous perspectives in order to have a thorough grasp of the factors that influence technology acceptability in the healthcare industry. As shown in Table 1, some studies even included the perspectives of participants who were not employed in the healthcare industry, such as prospective consumers and recipients of healthcare technologies, because they believed that these individuals also had a significant influence on the adoption of new technologies in the healthcare industry. In order to better illustrate the relationships and patterns related to barriers to technology adoption, Table 4 provides visual assistance for visualising the main conclusions from the reviewed studies.

The TAM theory is popular in technology acceptance research because it makes it easier for participants and researchers to interact with the study by reducing the complexity of the technology acceptance assessment to two key constructs: perceived usefulness and perceived ease of use. Nevertheless, it has demonstrated several significant limitations when applied to this industry due to the complexity of healthcare contexts. Therefore, researchers who utilise the TAM theory for the adoption of technology in the healthcare industry need to modify the TAM theory by adding additional variables or merging it with other theories in order to capture all relevant elements and provide trustworthy results. This is due to the fact that the TAM constructs of perceived usefulness and perceived ease of use are unable to adequately address the unique characteristics and complexity of healthcare settings. This is further demonstrated in this study, where every research paper that was included in the analysis demonstrated that, in order to account for every potential aspect, additional variables such as perceived trust and advantages to patient care had to be added. Based on the reviewed research, only three of the nine studies that were included in the review had a significant impact on the intention to use, suggesting that consumers’ motivation to adopt healthcare innovations was primarily unaffected by perceived usefulness. However, five of the nine studies demonstrated that perceived ease of use had a significant impact on the uptake of technology in the healthcare sector, making it a more crucial component. This is in contrast with the previous studies, which mainly focused on healthcare professionals and thus deemed usefulness to be more significant. The inclusion of patients, potential users, and beneficiaries in addition to healthcare personnel in the reviewed research may have an influence on this study’s conclusion, which led to the perceived ease of use being more significant.

The UTAUT theory is also a popular theory in healthcare theses because it includes four constructs that provide a structured and validated framework for understanding technology acceptance, which are performance expectancy, effort expectancy, social influence, and facilitating condition. These constructs offer an organised and verified framework for understanding technology acceptance. However, researchers still need to modify the UTAUT theory to fit the unique requirements of the healthcare industry even with this theory. This is due to the fact that the healthcare industry has a complex setting, which includes a wide range of stakeholders, including doctors, nurses, administrators, and patients, and each of the stakeholders has varying technology needs and acceptability standards. Additionally, there are other factors influencing technology adoption in the healthcare industry, such as complicated processes, concerns about data security, and perceived risks to patient safety. Because of these complexities, researchers using the UTAUT theory often need to adapt it by adding new variables or integrating it with other theories to cover all the possible factors and aspects. For example, variables such as perceived risk and trust should be included to meet the particular requirements of healthcare environments. This adaptability was demonstrated in this study, as few studies stick to the original UTAUT framework, while most alter it by adding external factors or mixing it with other theories. According to the reviewed research, behavioural intention was impacted by performance expectancy in 3 out of 10 studies. On the other hand, 4 out of 10 studies found that behavioural intention was influenced by social influence. Besides that, 3 out of 10 studies found that behavioural intention was influenced by effort expectancy.

According to this result, this study highlighted that the UTAUT theory is typically more appropriate for complex settings like healthcare settings, despite the TAM theory being more straightforward to apply than the UTAUT. This is due to the UTAUT theory’s comprehensive constructs and ability to take into account a more excellent range of variables and perspectives. Besides that, its adaptability allows it to have more accurate findings in these complicated settings, even without significant modifications. On the other hand, the TAM is more suitable for more straightforward settings like online commerce and e-learning adoption due to its more straightforward constructions as people interact with the technology directly and simply in these circumstances, making perceived usefulness and perceived ease of use highly related, and thus requiring less modification in these settings. To sum up, it is important for investigators to carefully consider the specific requirements and conditions of their healthcare settings when selecting the model that best fits their research.

From a theoretical standpoint, the study’s findings demonstrated that the TAM and UTAUT theories must be modified to account for the complexity of the healthcare setting when these two theories are applied in this industry. This can bias the results because the modified theories may be less relevant and effective in the healthcare setting. Although the evaluated research in this study used modified TAM or UTAUT theories, the dependability of each of the new variables was demonstrated by prior investigations. As such, the results of these studies may also be relied upon. Furthermore, theoretical improvements should focus on expanding the use of the UTAUT theory while also retaining the framework’s inherent strengths, as it has a comprehensive construct and is a reliable framework for studying technology adoption in complex settings such as healthcare.

From a practical standpoint, this study highlighted the importance of social influence, performance expectancy, effort expectancy, and perceived ease of use on technology adoption in the healthcare industry. Therefore, healthcare providers and associated staff should develop tailored strategies based on these variables in order to boost the rates of technology adoption in the industry. In addition, it is also essential to include stakeholders of the healthcare industry in the technology adoption process, such as patients and future users. This is due to the fact that their varied points of view may provide insightful information and contribute to the successful execution of healthcare technologies.

Additionally, social and organisational organisations, such as hospitals, should consider adopting an instructional strategy. This is due to the belief that education can affect an individual’s viewpoint about the use of technology [41]. For example, healthcare professionals believe that education can provide the instruction and training required to boost their confidence in technology. This may result in positive impacts on their perceived ease of use, performance expectations, and effort expectations of the use of new technologies, all of which may directly influence a shift in their conduct towards technology. On the other hand, educated medical professionals are better equipped to appraise technology and boost their peers’ faith in it, enabling the seamless adoption of new technologies, which may have a favourable impact on the societal influence of new technologies. At the same time, individuals’ willingness to use new technologies to adapt to the digital world will improve. Meanwhile, technical advancements in this field have the potential to enhance human growth. Specifically, this can enhance the professional growth of healthcare professionals to some extent through the use of technology. Healthcare professionals, for example, could reduce their workload while increasing their effectiveness thanks to modern technologies. Not only that, but medical care could improve as a result of novel technology, allowing for quick access to a patient’s medical history, medications, allergies, and test results and reducing errors caused by incomplete or inaccurate information.

From the patient’s standpoint, it is critical to increase public awareness and education about medical technological developments. By popularising the benefits and dependability of technology, patients can gain trust in technological innovations, improving the perceived ease of use and usefulness of new medical devices and thereby modifying behavioural intentions. This shift in attitude has the potential to significantly increase patients’ acceptance of new technologies while also boosting human development. For example, people with diabetes can now more readily check their blood sugar levels by using the latest technology, like continuous glucose monitoring (CGM) devices. This enables them to make prompt dietary or medication adjustments that improve their overall health management.

Thus, targeted teaching initiatives for both patients and healthcare providers have to be carried out. For instance, the organisation should establish peer mentoring or practical training to enable healthcare professionals to become familiar with the technology as soon as possible. With this, they would be able to ask for help if they run into any issues or technical difficulties while using the device. In order to foster trust and understanding among patients, social services should run public awareness campaigns or provide educational resources.

While tailored interventions, such as teaching initiatives and trust-building techniques, are essential for minimising barriers to the adoption of healthcare technology, implementing these interventions in different healthcare settings has unique challenges.

First, addressing user concerns about data security and transparency is critical to fostering trust in healthcare technologies. Therefore, governments should develop policies that promote trust, such as enforcing international data privacy laws such as GDPR, to enhance user confidence. For example, by ensuring openness in data use and consent protocols, opposition to the deployment of technologies such as teleconsultation and mobile health applications can be reduced [42]. Furthermore, offering financial incentives, such as grants for training initiatives and subsidies for the purchase of technology, can help with organisational readiness issues, particularly in environments with limited resources.

In addition, targeted training and organisational support are essential to encourage technology adoption to achieve practical implementation of medical technologies. The teaching initiatives must be specific to each role in order to satisfy the diverse requirements of healthcare professionals and patients. Healthcare professionals can benefit from role-specific training modules, which will further boost their confidence and enthusiasm towards applying new technologies. These positions include administrators, physicians, and nurses. For instance, simulation-based training can help emergency department employees become more adept at utilising clinical decision support systems [43], which can help with usability problems in high-stress situations. Moreover, it is important for companies to involve users, including healthcare professionals, in the technology development process. This ensures that the interface is user-friendly and adaptable to different levels of digital literacy. At the same time, continuing to integrate new technologies into existing workflows through feedback mechanisms and technical support can improve acceptability and operational efficiency.

To ensure that these strategies work in practice, small-scale pilot projects must be carried out to improve the interventions prior to widespread adoption. These procedures ensure that interventions are targeted and effectively address the unique challenges encountered in different healthcare settings. Technology in healthcare may be effectively incorporated by relating these customised treatments to actual circumstances. This will eventually lead to higher adoption rates and better patient outcomes.

However, further improvements of these strategies require strong research support. This analysis identifies critical areas for further research in order to enhance theoretical models and practical strategies. Although privacy and trust are often mentioned as important factors, little is known regarding the effectiveness of activities aimed at fostering trust. Future research should focus on assessing these treatments through longitudinal studies to ascertain how they impact technology adoption. Comparative studies conducted in different healthcare settings can uncover domain-specific barriers and facilitators, which will help create customised solutions. Additionally, expanding the TAM and UTAUT frameworks by incorporating behavioural, psychological, and organisational variables can provide a more comprehensive understanding of adoption dynamics.

In conclusion, addressing these research, practice, and policy issues may help bridge the gap between theoretical frameworks and real-world implementation.

### 4.3. Limitations and Future Research

This study has many restrictions. First off, there is not enough evaluated research to achieve results that are more thorough and precise. It is common knowledge that a larger body of research would enable a more detailed examination and more trustworthy outcomes. The number of reviewed studies is critical in conducting research and significantly contributes to the accuracy of the results. If the sample size is limited, there is a possibility that the results will be affected and might not accurately reflect the broad range of the healthcare industry.

Second, the study did not state in which area of the healthcare industry it was carried out. As is well known, the healthcare industry is a broad one with several areas, and many technologies may be used in each. However, the fact that this study was wide in its emphasis and did not target any particular areas raised the possibility of bias or erroneous findings. This is due to the possibility that different areas will yield different results from studies on technology adoption. For example, certain areas within the healthcare industry may only be relevant to healthcare professionals; in this instance, patient perspectives are not particularly relevant to the adoption of technology in this sector because patients will not be interacting with it.

Third, this study only employed the TAM and UTAUT frameworks and did not include newer or hybrid models like the Extended TAM (ETAM) or UTAUT-2, which may be more beneficial in specific areas as they include additional factors. The primary reason for excluding these new models is that, compared to the TAM and UTAUT, there are comparatively few related studies, which may not offer enough empirical evidence to support the findings, limiting their representativeness and undermining the review’s overall credibility. To guarantee the accuracy and comparability of the findings, this study employed the more established and popular TAM and UTAUT as its theoretical foundation.

Besides that, most of the studies included in this study were conducted in high-income and upper-middle-income countries, such as China, South Korea, Italy, the UAE, and so on. Therefore, the findings of this study may not apply to lower-middle-income and low-income countries, as they primarily reflect the healthcare environments of high-income and upper-middle-income countries. Although some studies include data from low-income countries such as India and Nigeria, the conclusions are mainly applicable to areas with sophisticated healthcare infrastructure. As a result, these findings have limited applicability to global healthcare systems, particularly in environments with minimal resources.

Consequently, future research should include a larger number of studies for review, such as those studies that used either the TAM, UTAUT or both theories in order to offer a broader range of views and contributing factors and to achieve a more profound knowledge of technology adoption in this intricate subject in order to increase the adoption rate. Furthermore, future research should widen the theoretical framework to incorporate new or hybrid models that make significant contributions, such as the Extended TAM (ETAM) and UTAUT-2, in order to achieve the most relevant and precise results. These modifications may incorporate other factors unique to healthcare environments, leading to more accurate and pertinent outcomes.

The suggested comprehensive solutions must be experimentally validated in order to increase the frameworks’ practical usefulness. This is due to the fact that the applicability and efficacy of these adjustments may be confirmed by carrying out research evaluating them in actual healthcare environments. In addition, due to the vast range of the healthcare industry, future research should focus on a specific area within the industry, such as the adoption of technology in emergency departments, to achieve more accurate and relevant results. By focusing on a specific area, researchers can gain a deeper understanding of the particular traits that affect the adoption of technology in that area, producing more precise and useful results.

Furthermore, future research should expand the scope of the study to include cultural and demographic diversity as important determinants. The findings will be more internationally relevant and applicable to worldwide healthcare settings if studies from various socioeconomic and geographic sources are combined. This diversity can help develop inclusive and successful strategies in many contexts by providing insights into how demographic and cultural aspects influence technology adoption.

In conclusion, future research should prove the utility of the theoretical frameworks and expand their application while also modifying them for specific settings. By addressing these issues, researchers may provide more comprehensive and globally relevant insights about the usage of healthcare technology.

### 4.4. Conclusions

By conducting this review, the TAM and UTAUT theories were found not to be designed to handle complex healthcare technologies, such as electronic health records or nursing systems, because they were first developed without considering crucial factors like organisational, cultural, and emotional implications [6]. Nevertheless, these elements significantly impact how technology is applied in the healthcare industry [6].

Despite these limitations, the TAM and UTAUT still have gained popularity in healthcare due to their straightforward technology acceptance assessments and simplified processes. This is because both offer solid theoretical foundations for comprehending the adoption of technology. Still, their ability to effectively effect significant change hinges on how well they can adjust to the unique constraints faced by the healthcare industry. In order to make these theories more applicable to the healthcare sector and produce more accurate findings, researchers have had to enhance the models by including outside variables or keeping the models environment-appropriate.

Additionally, this study indicated that the benefit of the TAM is its simplicity, which makes it easier to incorporate. Nevertheless, it must be modified to include external variables in order to handle complex environments. However, the modified TAM has been shown to be able to adapt to complex environments. In contrast, the UTAUT has a comprehensive framework that covers a broader range of factors, such as technological, psychological, social, and environmental dimensions, which makes it more adaptable in complex environments. However, due to the complexity of the framework, the implementation of the UTAUT usually requires a lot of expertise and resources, and modifications to its framework are difficult to avoid, affecting its effectiveness. Nevertheless, even without major modifications, the UTAUT can adapt to environments that are not too complex.

In summary, even though the TAM theory might struggle with the intricacies of healthcare settings, the UTAUT theory may be used in a healthcare-related context if the study environment is not overly complicated. This result further suggested that the UTAUT theory might be more appropriate for studies of the healthcare industry under certain conditions. This is because the UTAUT encompasses more aspects than the TAM, such as technology, psychological, social, and environmental factors [11]. In conclusion, although the TAM and UTAUT theories may not provide a comprehensive understanding of the intricacy involved in new technologies in the healthcare sector, they can still offer dependable and consistent prediction abilities for the adoption and utilisation of technology in healthcare environments by incorporating additional variables. However, as the UTAUT may be applied without modification in simple healthcare-related research settings, including those in the healthcare sector, it may be argued that it better suits the needs of healthcare-related research.

Apart from that, the results of this review indicated that the psychological, social, and organisational factors are the most important ones that organisations, such as hospitals and society, should concentrate on. The finding of this study suggested that organisations, such as hospitals, should offer thorough education and training programmes that include scenario-based instruction, phased training plans, personalised training materials, peer and expert collaboration, ongoing support and resources, and so on. The adoption and confidence of healthcare providers in technology can be successfully increased by putting these strategies into practice. For example, organisations should specifically offer technology operation training that mimics real-world work situations so that medical professionals can experience the benefits and use of technology in clinical settings. Simultaneously, they must create a methodical learning route that progressively moves from fundamental technology usage abilities to sophisticated operations in order to guarantee that every employee has an adequate understanding of technology. Moreover, organisations should also design training materials according to the duties and technical requirements of various roles. For instance, nurses’ training should focus more on technology that relates to patient care, whereas doctors’ training should concentrate on technology that relates to diagnosis and decision-making. Additionally, hospitals can also arrange some peer-sharing sessions or invite technical experts to conduct professional training to increase the confidence and trust of medical professionals through experience sharing. In addition, they should also establish a technical support staff to respond to enquiries and offer assistance whenever needed, as well as offer easily accessible online learning materials such as system operating manuals, technical guides, and operation videos. By using these multi-layered and diverse education strategies, organisations can assist medical professionals in using technology more successfully, lower resistance to adoption, and enhance the overall quality of treatment.

In addition to organisations’ efforts, society at large should focus more on medical technology education for the general population. For instance, popular scientific events like talks, exhibits, or hands-on activities can be held to educate the public about the capabilities and benefits of new medical technologies at the local level. Through these events, the general public may experience the benefits that technology offers and gain an understanding of how it might enhance patient health and medical efficiency understandably. Furthermore, medical technology application instances can be disseminated via social media and television shows, and more people can be made aware of the technology’s potential and worth through brief films or special reports. In order to allow the public to gradually understand medical technology, information about health management software, medical robots, and so on should be included in primary and secondary school and university courses to improve basic medical technology knowledge. Meanwhile, in order to enable students to have a deeper understanding of medical technology in practice, competitions can also be organised to stimulate their interest. Through comprehensive social education and promotion, it is possible to enhance public acceptability of medical technology and encourage its extensive use across all levels.

It is believed that these strategies can successfully improve the acceptance of medical technologies by both medical professionals and patients from a psychological perspective by reducing psychological resistance and enhancing confidence. Medical professionals and patients can enhance or even speed up the acceptance and use of medical technology by lowering their fear about it and gradually cultivating a positive attitude through organisation-provided training, public awareness campaigns, and education in society. At the same time, the extensive use of medical technology can simultaneously improve public health and make healthcare more accessible. Technologies like telemedicine, for instance, can help address the issue of inadequate medical resources in rural areas and give more people access to equal medical possibilities. In order to accomplish these goals, it is not only required to increase healthcare professionals’ and society’s psychological acceptance of medical technology but also to combine specific tactics to ensure effective technology adoption. For instance, it may efficiently lessen the operational challenges faced by healthcare professionals and facilitate their quicker adoption of new technologies by streamlining the technology’s operating interface and offering real-time technical assistance. By expanding access to personalised services and medical technologies, society’s trust in technology can be enhanced. For example, society’s adoption of medical technology can be improved by personalised health management apps that offer recommendations based on individual health information. Furthermore, healthcare providers may benefit from the expertise of other industries, such as the customer feedback system in the retail industry. By establishing the customer feedback system, they will be able to create channels for constantly obtaining user feedback and immediately adjusting the features and services of technology in order to better the user experience. These have the potential to improve the quality and efficacy of medical treatment, encourage wider technology adoption, and increase public and healthcare professionals’ trust in developing technologies.

## Figures and Tables

**Figure 1 healthcare-13-00250-f001:**
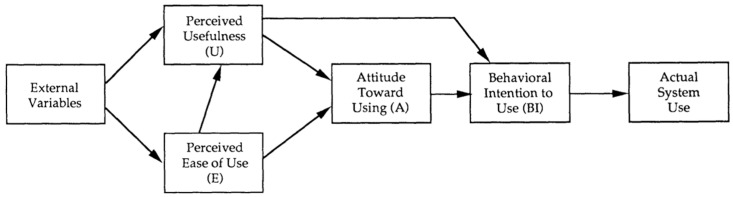
Technology Acceptance Model (TAM) adapted from Davis et al. (1989) [18].

**Figure 2 healthcare-13-00250-f002:**
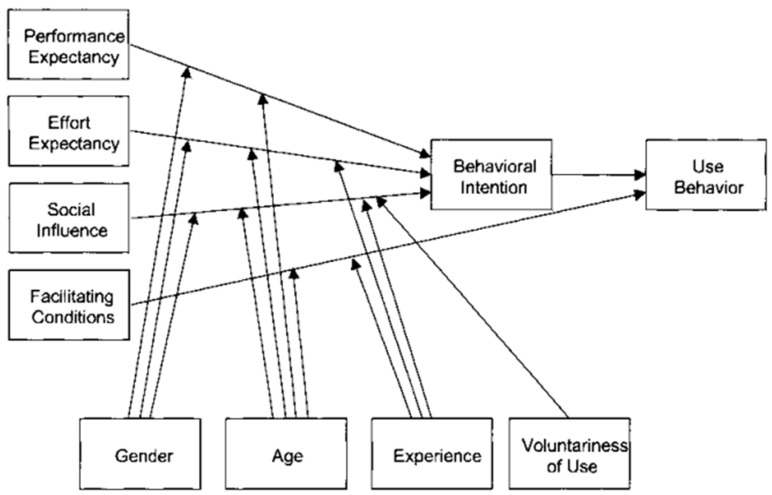
Unified Theory of Acceptance and Use of Technology (UTAUT) adapted from Venkatesh et al. (2003) [21].

**Figure 3 healthcare-13-00250-f003:**
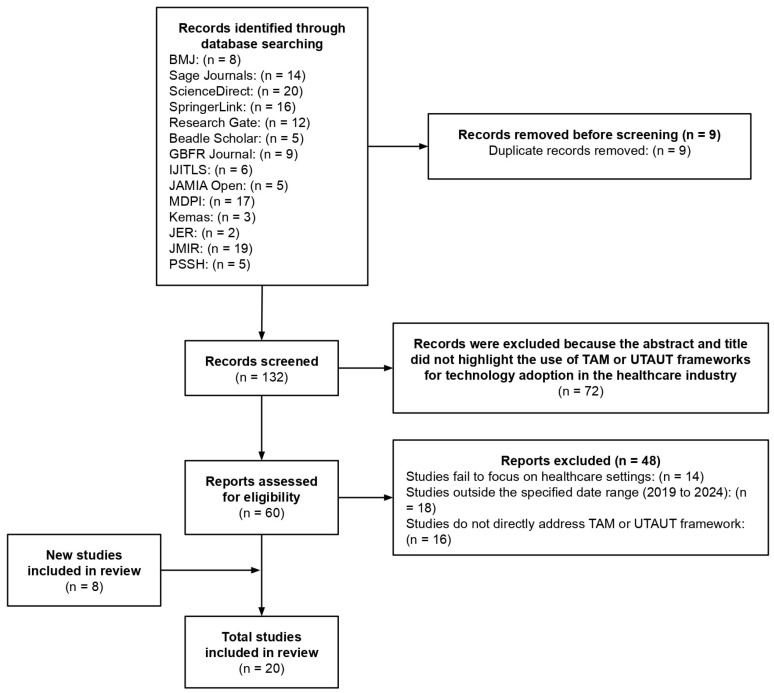
PRISMA flow diagram of the study selection process.

**Table 1 healthcare-13-00250-t001:** Summary of reviewed studies, topics, participants, and sample sizes.

Authors	Title	Participants	Sample Size
Zoccarato et al. (2024) [12]	Unveiling the interplay between rational, psychological and functional factors in continuous glucose monitoring early adoption: Novel evidence from the Dexcom ONE case in Italy	Users of Dexcom ONE blood sugar monitors in Italy	157 responses
Ekaimi et al. (2024) [5]	Examining the Factors Influencing Teleconsultation Adoption During the Pandemic Using the TAM Model	Indonesian patients aged 20–40 with a bachelor’s degree who used teleconsultation services during the COVID-19 pandemic	100 responses
Zin et al. (2023) [24]	A Study on Technology Acceptance of Digital Healthcare among Older Korean Adults Using Extended Tam (Extended Technology Acceptance Model)	Elderly Koreans (56 years and older) from Busan using digital healthcare tools	170 responses
Alhashmi et al. (2019) [25]	Implementing Artificial Intelligence in the United Arab Emirates Healthcare Sector: An Extended Technology Acceptance Model	IT and medical staff from 13 health facilities in Dubai	53 responses
Nazari-Shirkouhi et al. (2023) [26]	A model to improve user acceptance of e-services in healthcare systems based on technology acceptance model: an empirical study	Patients who used e-services in hospitals over the past year	216 responses
Dhagarra et al. (2020) [13]	Impact of Trust and Privacy Concerns on Technology Acceptance in Healthcare: An Indian Perspective	Indian patients	416 responses
Walle et al. (2023) [27]	Predicting healthcare professionals’ acceptance towards electronic personal health record systems in a resource-limited setting: using modified technology acceptance model	Medical professionals from Amhara regional state teaching hospitals in Ethiopia	638 responses
Wang et al. (2023) [28]	Predicting pediatric healthcare provider use of virtual reality using a technology acceptance model	Paediatric healthcare professionals and non-healthcare clients	270 responses
Karkonasasi et al. (2023) [29]	Acceptance of a Text Messaging Vaccination Reminder and Recall System in Malaysia’s Healthcare Sector: Extending the Technology Acceptance Model	Nurses from Malaysian government hospitals or clinics	128 responses
Nurhayati et al. (2019) [31]	Unified Theory of Acceptance and Usage of Technology (UTAUT) Model to Predict Health Information System Adoption	Nutrition officers	50 responses
Osifeko et al. (2019) [22]	A Modified Unified Theory of Acceptance And Use of Technology (Utaut) Model For E-Health Services	Users of e-health services at ten medical facilities in Lagos, Nigeria	210 responses
Zhou et al. (2019) [30]	Assessment of the social influence and facilitating conditions that support nurses’ adoption of hospital electronic information management systems (HEIMS) in Ghana using the unified theory of acceptance and use of technology (UTAUT) model	Nurses at five major public hospitals in Ghana	660 responses
Zhu et al. (2023) [14]	Understanding Use Intention of mHealth Applications Based on the Unified Theory of Acceptance and Use of Technology 2 (UTAUT-2) Model in China	General respondents via a professional survey platform	371 responses
Philippi et al. (2021) [32]	Acceptance towards digital health interventions—Model validation and further development of the Unified Theory of Acceptance and Use of Technology	Secondary analysis from 14 papers, with 10 providing primary data	Data from 10 studies with primary data
Farhady et al. (2020) [33]	Evaluation of effective factors in the acceptance of mobile health technology using the unified theory of acceptance and use of technology (UTAUT), case study: Blood transfusion complications in thalassemia patients	Haematologists	58 responses
Yin et al. (2022) [35]	User acceptance of wearable intelligent medical devices through a modified unified theory of acceptance and use of technology	Respondents via the WeChat app survey	2192 responses
Yousef et al. (2021) [36]	Predicting Patients’ Intention to Use a Personal Health Record Using an Adapted Unified Theory of Acceptance and Use of Technology Model: Secondary Data Analysis	Users of the MNGHA Care PHR (personal health record)	324 responses (secondary analysis from original 546)
Wrzosek et al. (2020) [38]	Doctors’ Perceptions of E-Prescribing upon Its Mandatory Adoption in Poland, Using the Unified Theory of Acceptance and Use of Technology Method	Primary care physicians in Poland’s private sector	381 responses
Christian et al. (2023) [39]	Generation YZ’s E-Healthcare Use Factors Distribution in COVID-19’s Third Year: A UTAUT Modeling	Generations Y and Z in Indonesia	268 responses (134 from Generation Y and 134 from Generation Z)
Edo et al. (2023) [40]	Why do healthcare workers adopt digital health technologies—A cross-sectional study integrating the TAM and UTAUT model in a developing economy	Healthcare professionals	125 responses

**Table 2 healthcare-13-00250-t002:** Modifications and applications of theoretical frameworks in reviewed studies.

References	TAM	UTAUT	UTAUT-2	Revised TAM	Revised UTAUT	Revised UTAUT-2	Remark
Zoccarato et al. (2024) [12]				/			Added external variables
Ekaimi et al. (2024) [5]				/			Added external variables
Zin et al. (2023) [24]				/			Added external variables
Alhashmi et al. (2019) [25]				/			Added external variables
Nazari-Shirkouhi et al. (2023) [26]				/			Added external variables
Dhagarra et al. (2020) [13]				/			Added external variables
Walle et al. (2023) [27]				/			Added external variables
Wang et al. (2023) [28]				/			Added external variables
Karkonasasi et al. (2023) [29]				/			Added external variables
Nurhayati et al. (2019) [31]		/					-
Osifeko et al. (2019) [22]					/		Added external variables
Zhou et al. (2019) [30]					/		Removed some variables
Zhu et al. (2023) [14]						/	Added external variables
Philippi et al. (2021) [32]					/		Added external variables
Farhady et al. (2020) [33]					/		Added external variables
Yin et al. (2022) [35]					/		Combined multiple theories
Yousef et al. (2021) [36]					/		Model adaptated and additional variables added
Wrzosek et al. (2020) [38]		/					-
Christian et al. (2023) [39]				/	/		Added external variables
Edo et al. (2023) [40]				/	/		Integrated two theories and added external variables

**Table 3 healthcare-13-00250-t003:** Comparison table of TAM theory and UTAUT theory.

**Methodology**	**TAM**	**UTAUT**
Advantages	Provides a solid foundation to support empirical research. Simplifies the assessment of technology acceptability. Uses two key constructs for speedy implementation and analysis. Highly flexible and adaptable.	Effectively used in numerous study fields with substantial empirical support across industries. Comprehensive framework enables direct or minor modification for application to a complex setting. Four main components offer an integrated perspective on technology adoption and provide a more accurate understanding of various factors influencing technology adoption.
Limitations	May overlook factors affecting technology adoption in complex settings as it is oversimplified. Requires inclusion of additional variables for precise conclusions in complex settings.	Difficult to use and comprehend due to its comprehensive framework. Requires researchers to have a solid grasp of the theory because of its complexity. Takes a long time for application due to the need for careful implementation. Less adaptable in terms of modification to particular settings without compromising theoretical integrity.

**Table 4 healthcare-13-00250-t004:** Key barriers and insights from reviewed studies.

References	Framework Used	Barriers Identified	Key Findings
Zoccarato et al. (2024) [12]	TAM	Trust, functional usability	Rational factors significantly impact CGM adoption by diabetic patients.
Ekaimi et al. (2024) [5]	TAM	Trust, privacy, lack of training	Trust and privacy strongly influence teleconsultation adoption.
Zin et al. (2023) [24]	Extended TAM	Digital literacy, stigma	Perceived ease of use and trust influence elderly Koreans’ adoption of digital health tools.
Alhashmi et al. (2019) [25]	Extended TAM	Organisational, strategic factors	Managerial and organisational support are critical for AI adoption in UAE healthcare.
Nazari-Shirkouhi et al. (2023) [26]	TAM	Service quality, user satisfaction	High-quality interfaces and positive user experiences improve e-service adoption in healthcare.
Dhagarra et al. (2020) [13]	Revised TAM	Trust, privacy concerns	Trust promotes adoption, while privacy concerns hinder technology use in India.
Walle et al. (2023) [27]	Revised TAM	Digital literacy, IT experience	Healthcare professionals’ digital literacy affects ePHR adoption.
Wang et al. (2023) [28]	Revised TAM	Age, perceived enjoyment	Virtual reality adoption in paediatric healthcare is influenced by perceived ease of use and enjoyment.
Karkonasasi et al. (2023) [29]	Extended TAM	System compatibility, privacy concerns	Compatibility positively impacts nurses’ attitudes toward VHC.
Nurhayati et al. (2019) [31]	UTAUT	Effort expectancy, resource availability	Effort expectancy significantly impacts health information system adoption.
Osifeko et al. (2019) [22]	Modified UTAUT	Social influence, ICT infrastructure	Social influence is the strongest predictor of e-health adoption in Nigeria.
Zhou et al. (2019) [30]	UTAUT	Social influence, resource availability	Social influence critically affects nurses’ use of HEIMSs in Ghana.
Zhu et al. (2023) [14]	UTAUT-2	Price sensitivity, risk perception	Perceived trust and performance expectancy drive mHealth adoption.
Philippi et al. (2021) [32]	UTAUT	Internet anxiety, data security	Social support mitigates anxiety and boosts digital health adoption.
Farhady et al. (2020) [33]	Modified UTAUT	Accessibility, social/technological barriers	Reliable mHealth technology and robust accessibility features reduce adverse outcomes.
Yin et al. (2022) [35]	Modified UTAUT	Cost, health expectations	Health expectations and social support influence WIMD adoption.
Yousef et al. (2021) [36]	Revised UTAUT	Attitude, performance expectancy	Positive attitudes strongly predict PHR adoption in Saudi Arabia.
Wrzosek et al. (2020) [38]	UTAUT	Standardisation, training gaps	Custom interfaces and training improve e-prescribing adoption in Poland.
Christian et al. (2023) [39]	UTAUT	Effort expectancy, generational differences	Perceived ease of use significantly impacts adoption.
Edo et al. (2023) [40]	Integrated TAM-UTAUT	Trust, user experience	Integration of frameworks highlights the role of trust in digital health adoption.

## Data Availability

This study does not create ethical issues about data security, privacy, or confidentiality because of the nature of the review and the lack of human participants in the research process.

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
