# Peer review of "Understanding Psychosocial Barriers to Healthcare Technology Adoption: A Review of TAM Technology Acceptance Model and Unified Theory of Acceptance and Use of Technology and UTAUT Frameworks"

_healthcare, 2025, doi:10.3390/healthcare13030250_

Round 1
Reviewer 1 Report
Comments and Suggestions for Authors
The manuscript explores the application of established frameworks, namely the TAM and the UTAUT, to analyze barriers to healthcare technology adoption. While the review addresses a pertinent topic, there are several areas that would benefit from further elaboration and refinement to enhance the overall clarity, depth, and impact of the study.
The abstract provides a concise overview of the research objectives and methods, but its discussion of key findings lacks precision in presenting the practical implications of the results. A stronger emphasis on actionable insights for healthcare practitioners or policymakers would improve the abstract’s utility. The claim that future research should explore integrative frameworks is valid but would benefit from more concrete suggestions on what such frameworks might include.
The introduction effectively sets the stage by emphasizing the importance of healthcare technology in enhancing outcomes. However, its review of existing literature misses an opportunity to critically analyze the gaps that justify the need for this study. The discussion of TAM and UTAUT appears somewhat general, with limited exploration of how these frameworks have been adapted or extended in specific healthcare contexts (10.1016/j.jjimei.2023.100186, 10.1016/j.ijhcs.2022.102922). Incorporating a more detailed critique of how current adaptations of these models fall short in addressing domain-specific challenges would add depth to the rationale.
The methods section demonstrates a systematic approach, particularly in its screening and selection criteria for reviewed articles. The focus on recent literature enhances the relevance of findings, but the inclusion process could be better justified by explaining how the chosen studies comprehensively represent the field. Additionally, while the use of narrative synthesis is appropriate given the data constraints, further details on how qualitative and quantitative insights were reconciled would strengthen the methodological rigor.
The results section organizes findings well, highlighting key barriers to technology adoption, such as trust and organizational readiness. Nevertheless, the presentation relies heavily on textual summaries, which makes it harder to draw connections between different studies. Visual aids, such as comparative tables or thematic maps, could greatly enhance the accessibility and interpretability of findings. The assertion that UTAUT is more comprehensive than TAM is valid but overly simplistic, as the nuanced trade-offs between these frameworks are not fully explored.
The discussion highlights the need for tailored interventions but does not delve deeply enough into the complexities of implementing such strategies in healthcare settings. For instance, while the importance of trust and education is mentioned, practical approaches for fostering these elements in diverse healthcare environments remain underexplored (10.1080/10447318.2024.2410536, 10.13140/RG.2.2.28353.33126). There is an opportunity to connect the discussion more explicitly to real-world scenarios, such as specific case studies or pilot programs.
The conclusion aligns well with the findings, emphasizing the potential of modified TAM and UTAUT frameworks to address adoption barriers. However, it would benefit from a more definitive stance on whether these frameworks, in their current or adapted forms, are sufficient for driving meaningful change. Offering specific recommendations for stakeholders would enhance the conclusion’s practical value.
The manuscript’s discussion of ethical considerations is minimal, and it is unclear whether the reviewed studies uniformly adhered to ethical standards. A broader acknowledgment of ethical implications, particularly concerning trust and data privacy in healthcare technology, would be valuable.
The writing is generally clear and accessible, but there are instances where overly technical or repetitive phrasing detracts from readability. The review would benefit from a more polished narrative that succinctly integrates key findings while maintaining academic rigor.
The study’s contribution to the field is significant, given its focus on behavioral and organizational factors that influence technology adoption. However, its novelty is somewhat diluted by the lack of concrete examples and actionable recommendations (10.1186/s12909-024-05680-z, 10.1145/3544548.3580682). The review would be more impactful if it went beyond synthesizing existing literature to propose new avenues for research or practice.
In summary, the manuscript presents a well-structured and relevant review but falls short in several critical areas, including the depth of analysis, methodological transparency, and practical application. Addressing these gaps through minor revisions would substantially enhance its value to the field. Accepting the manuscript with revisions aimed at improving these aspects is recommended.
Comments on the Quality of English LanguageThe English could be improved to express the research more clearly. While the text is understandable, some sections include repetitive or overly technical phrasing that detracts from readability.
Reviewer 2 Report
Comments and Suggestions for Authors
Dear Authors,
I hope this message finds you well. Thank you for submitting your manuscript, "Understanding Psychosocial Barriers to Healthcare Technology Adoption: A Review of TAM and UTAUT Frameworks," for consideration. Your focus on the psychosocial barriers to healthcare technology adoption and the integration of TAM and UTAUT frameworks addresses an important and timely topic. After a thorough review, I would like to provide feedback on the manuscript. While the subject matter is undoubtedly significant, there are several key issues that limit the contribution of the manuscript to the field, which may affect its suitability for publication in its current form.
Strengths of the Manuscript
The topic is highly relevant, particularly as healthcare systems increasingly integrate technology. The structured review of TAM and UTAUT frameworks offers a clear comparison and highlights the potential for adaptation in healthcare settings. Your emphasis on behavioral and psychological factors adds a valuable perspective to the discourse on technology adoption.
Areas of Concern
- Limited Scope of Frameworks: The review focuses exclusively on TAM and UTAUT, overlooking newer or hybrid theoretical models that might offer more nuanced insights. Incorporating or at least acknowledging alternative models could broaden the manuscript’s applicability.
- Contextual Limitations: The reviewed studies primarily represent specific geographic and cultural contexts. This restricts the generalizability of the findings to global healthcare settings, which often vary significantly in terms of infrastructure, policies, and cultural attitudes.
- Underexploration of External Factors: External variables, such as regulatory barriers, resource limitations, and systemic healthcare challenges, are only lightly addressed. These factors often play a critical role in the real-world adoption of healthcare technology.
- Temporal Scope: The manuscript limits its review to articles published between 2019 and 2024. While this ensures recent data, it excludes earlier foundational research that might provide essential context or long-term trends.
These limitations collectively reduce the manuscript’s ability to offer a substantial advancement in understanding healthcare technology adoption. While the findings are relevant, they do not provide new or transformative insights that significantly advance existing literature. Without addressing these gaps, the manuscript risks being perceived as reiterative rather than innovative.
Recommendations
I recommend the following points to revise your manuscript:
- Expanding the theoretical frameworks reviewed to include emerging models or interdisciplinary approaches.
- Diversifying the geographic and cultural contexts of the included studies or explicitly discussing how the findings may vary across different settings.
- Examining systemic and external factors more comprehensively to present a holistic understanding of the barriers to healthcare technology adoption.
- Considering a broader temporal scope to include foundational studies that contextualize recent developments.
Comments on the Quality of English Language
The quality of the English language in the manuscript is adequate for academic writing but could benefit from improvement in the following areas:
1. Clarity and Precision: Some sentences are overly long and complex, making the arguments less clear. Simplifying sentence structures and ensuring concise expression would improve readability.
2. Grammar and Syntax: There are occasional issues with subject-verb agreement, article usage, and awkward phrasing. A thorough proofreading by a native or professional academic editor would help address these issues.
3. Consistency in Terminology: Key terms such as "healthcare technology adoption" and "psychosocial barriers" should be used consistently throughout the manuscript to avoid confusion.
4. Academic Tone: While the tone is mostly professional, there are sections where the language feels informal or conversational. These should be revised to maintain a consistent scholarly tone.
Reviewer 3 Report
Comments and Suggestions for Authors
The authors present an important review on "Understanding psychosocial barriers to the adoption of health technology: a review of the TAM and UTAUT frameworks." This study is considered current and of scientific importance for this MDPI journal.
To improve the quality of the manuscript, the authors must clarify the following:
1. Explain why aspects related to the educational level and prior knowledge (training) are not considered in the study. That is, because it may be intuitive that prior knowledge increases the use of health technology.
2. Explain why access barriers to technology and the digital divide are not analyzed in this study.
3. What is the objective of the study?
4. When applying PRISMA, it is vital to define: What are the research questions?
5. According to the results obtained: there are organizational obstacles. BUT it is important to contextualize the social and economic environment of the analyzed article. Organizations vary according to: market niche, competition, social environment, financial capacity, etc.
6. Explain how trust in healthcare technology is measured?
7. What adjustments were made for both the TAM and the UTAUT to understand the adoption of healthcare technology?
8. Finally, it is important that the authors present alternative solutions to the factors that impede the adoption of healthcare technology.
Round 2
Reviewer 2 Report
Comments and Suggestions for Authors
Key Areas for Improvement:
1. Abstract:
Issue: The abstract is densely packed with details, making it less accessible to a broader audience.
Recommendation: Simplify the language and focus on summarizing the key objectives, findings, and implications.
2. Introduction:
Issue: While the introduction provides a good overview, it lacks a strong hook or justification for the study's importance.
Recommendation: Enhance the introduction with a compelling statement or real-world example highlighting the urgency of addressing healthcare technology adoption barriers.
3. Literature Review:
Issue: The literature review is thorough but overly descriptive, summarizing studies without sufficient critical analysis.
Recommendation: Focus on synthesizing the findings and identifying gaps in the existing research to build a stronger foundation for the study.
4. Figures and Tables:
Issue: Figures like the PRISMA flowchart are useful but lack detailed captions or descriptions.
Recommendation: Ensure all visuals are self-explanatory and provide sufficient context.
5. Discussion:
Issue: The discussion section occasionally reiterates findings without delving into deeper implications or practical applications.
Recommendation: Expand on how the results can inform policy, healthcare practices, or future research.
6. Language and Style:
Issue: Some sentences are verbose, making them harder to follow.
Recommendation: Use concise language and avoid unnecessary jargon to enhance readability.
7. References:
Issue: The reference list appears comprehensive but includes some potentially outdated sources.
Recommendation: Ensure that more recent studies (from 2023–2024) are adequately represented to reflect current trends.
Suggestions for Future Work:
Explore how these frameworks could be adapted or combined with emerging models to address domain-specific challenges.
Conduct empirical validation of proposed integrative strategies to enhance their practical utility.
Include demographic and cultural diversity as key variables to make findings globally relevant.
